# Scaling Laws for Robust Comparison of Open Foundation Language-Vision Models and Datasets

**Marianna Nezhurina**[1,2,5§*]    **Tomer Porian**[1,2,5*]    **Giovanni Puccetti**[3]    **Tommie Kerssies**[1,4]
**Romain Beaumont**[1]    **Mehdi Cherti**[1,2,5§°*]    **Jenia Jitsev**[1,2,5°*]
[1]LAION    [2]Juelich Supercomputing Center (JSC), Research Center Juelich (FZJ)
[3] Institute of Information Science and Technologies "A. Faedo" - CNR Pisa
[4] Eindhoven University of Technology
[5] Open-Ψ (Open-Sci) Collective
§ equal first, ° equal senior, * core contribution
{m.nezhurina,m.cherti,j.jitsev}@fz-juelich.de,contact@laion.ai

## Abstract

In studies of transferable learning, scaling laws are obtained for various important foundation models to predict their properties and performance at larger scales. Taking language-vision learning as example, we show here how scaling law derivation can also be used for model and dataset comparison, allowing to decide which procedure is to be preferred for pre-training. Full scaling laws based on dense measurements across a wide span of model and samples seen scales are derived for two important language-vision learning procedures, CLIP and MaMMUT, that use either contrastive only or contrastive and captioning text generative loss. For the first time, we use derived scaling laws to compare both models and three open datasets, DataComp-1.4B, Re-LAION-1.4B and DFN-1.4B, while ensuring sufficient prediction accuracy on held out points. From comparison, we obtain evidence for (i) MaMMUT's stronger improvement with scale and better sample efficiency than standard CLIP (ii) DFN-1.4B outperforming other open datasets. To strengthen validity of the comparison, we show scaling laws for various downstream tasks, classification, retrieval, and segmentation, observing consistently the same scaling trends for models and datasets across tasks. We show that comparison can also be performed when deriving scaling laws with a constant learning rate schedule, reducing compute cost. Accurate derivation of scaling laws provides thus means to perform model and dataset comparison on aligned common compute axis across large scale span, avoiding misleading conclusions based on measurements from few isolated single reference scales only. This paves road for guided collective improvement of open foundation models and training datasets, as scaling law based comparisons from various studies executed in common frame can be combined to identify overall better procedures. We release all the pre-trained models with their intermediate checkpoints, including openMaMMUT-L/14, which achieves $80.3\%$ zero-shot ImageNet-1k accuracy, trained on 12.8B samples from DataComp-1.4B[1].

## 1 Introduction

Foundation models [1] pre-trained on generic, diverse large datasets enabled massive improvements in transfer learning, showing strong and efficient adaptability across a wide range of downstream tasks not shown during pre-training. Thanks to learning procedures leading to foundation models,

---

[1]Code for reproducing experiments in the paper and raw experiments data can be found in the repository

39th Conference on Neural Information Processing Systems (NeurIPS 2025).

transferable learning can be studied across various important domains, including language [2, 3, 4], vision [5], language-audio [6], and language-vision [7].

Progress in improving learning procedures leading to stronger foundation models crucially depends on the ability to perform systematic and consistent learning procedure comparison. Usually, following the training, various foundation models are compared via performance shown on a wide range of standardized reference downstream tasks. Often, this comparison is done on only one or few selected reference model and data scales, without carefully aligning the compute put into pre-training. Further, important parts of learning procedure like training dataset often remain closed. This makes it hard or impossible to discern whether the observed model differences are due to algorithmic, dataset or due to differences in pre-training compute, or a combination of them, leaving also unclear whether the comparison holds for other scales than the few selected ones.

In our work, we make a step towards resolving these issues by using scaling law derivation to enable systematic training procedure, model, and dataset comparison. Foundation models exhibit scaling laws [8, 9, 10, 11] that allow to determine dependence of model properties and performance on total pre-training compute from measurements on smaller scales, enabling predictions across a wide scale span instead of only one or few selected points. Here, we show how using open datasets for scaling law derivation can enable model and dataset comparison that takes into account model behavior across a wide range of pre-training compute budgets and across different downstream tasks, while offering full control over variations in the whole training pipeline, and being fully reproducible.

We choose language-vision learning as an important setting for model and dataset comparison using scaling law derivation. Contrastive language-image pre-training (CLIP) [7] is a well-established learning procedure resulting in models that show impressive robustness and transfer capability, and are used routinely as pre-trained components in many setups such as vision-language instruction tuning models (LLaVa [12], InternVL [13], SigLIP [14]) and text to image generation models [15]. Since the first release of CLIP, many extensions have been proposed such as CoCa [16], MaMMUT [17], and SigLIP [18, 14]. These works claim more performant language-vision models than standard CLIP. However, as hinted above, it is still unclear which of the training procedures are better for what reasons, and whether claims of improving on the standard CLIP procedure hold across scales. Here, we conduct a large-scale study of the scaling laws of two important procedures, namely CLIP and MaMMUT, pre-trained on open reference datasets, DataComp-1.4B [19], DFN-1.4B [20] and Re-LAION-1.4B [21], which we are able to compare via full scaling law derivation for the first time.

Our study reveals that while CLIP has advantage on smaller compute scales, MaMMUT architecture shows advantage as we increase compute, as illustrated by cross-over of the scaling curves in Fig. 1, 2,3 (Sec. 3.1). Importantly, we find that the comparison via scaling laws delivers result that is consistent and robust across the pre-training datasets, learning schedules and the downstream tasks.

**Our Contributions. 1)** We conduct first large-scale study of CLIP [7, 10] and MaMMUT [17] with dense measurements to ensure better prediction to unseen scales and valid model and dataset comparison. Measurements cover model sizes from S/32 to H/14, samples seen from 1.28M to 3B, training on DataComp-1.4B [19], Re-LAION-1.4B [21] and DFN-1.4B [20] datasets and evaluating downstream performance on tasks covering zero-shot classification, retrieval, and segmentation. **2)** Based on derived scaling laws, we perform model and dataset comparison. We show validity of this comparison by revealing consistent trends in favor of MaMMUT versus CLIP architecture across different downstream tasks and pretraining datasets, as well as distinct scaling law types (compute-optimal and samples seen based scaling laws). We also show consistent trends favoring DFN-1.4B over other datasets. Our study is thus the first to provide a fully reproducible systematic comparison of important open foundation models openCLIP and openMaMMUT and important reference open datasets Re-LAION-1.4B, DataComp-1.4B and DFN-1.4B. **3)** We perform the study using both cosine and constant learning rate schedules, and observe the same consistent trends, revealing that the scaling laws based comparison can be also performed with suboptimal constant learning rate schedule resulting in 98% less compute cost. **4)**. We provide a detailed procedure to fit scaling laws tailored for model and dataset comparison, sharing our findings on best practices of scaling law derivation which include uncertainty quantification and measuring held-out points on the compute or samples-seen axis. **5)** We make our study fully reproducible and release all the intermediate checkpoints from scaling law derivation experiments. For the first time, we provide an open-source implementation of MaMMUT, openMaMMUT, and release openMaMMUT-L/14 trained on 12.8B image-text samples from an open dataset DataComp-1.4B, achieving 80.3% zero-shot ImageNet-1k accuracy.

# 2 Methods & experimental setup

Our experiments entail model pre-training, evaluation and scaling law derivation, which we describe in the following.

## 2.1 Pre-training setup

**Architecture details**. The MaMMUT model is integrated into the `openCLIP` code base [22] to take advantage of the existing implementation of the CLIP contrastive loss and CoCa captioning loss. Key additions to implement MaMMUT were: (1) using a double forward pass for the text component to perform both text encoding (without masking) and decoding (with causal masking); (2) cross-attention between image and text tokens. For an N-layer text decoder, cross-attention layers are inserted every 2 text decoder layers [17], a total of $\left\lceil \frac{N}{2} \right\rceil$ cross-attention layers.

**Training dataset & objective**. We pre-trained CLIP and MaMMUT models on DataComp-1.4B [19], Re-LAION-1.4B [21] and DFN-1.4B [20] datasets. Re-LAION-1.4B was obtained by downloading the `Re-LAION-2B-en-research` subset of Re-LAION [21] which contained $\approx 30\%$ dead links, resulting in a total of 1.4B image-caption pairs. DFN-1.4B was obtained by downloading DFN-2B dataset [20], which also resulted in discovering $\approx 30\%$ link rot, again providing 1.4B image-text pairs in total. CLIP models are trained with contrastive InfoNCE [23] loss ($L = L_{\text{contrastive}}$), while MaMMUT models are trained with contrastive and captioning losses ($L = L_{\text{contrastive}} + \lambda L_{\text{cap}}$), we used $\lambda = 1$ in our experiments.

**Model & samples seen scales**. For both CLIP and MaMMUT architectures, we consider ViT-S, ViT-M, ViT-B, ViT-L, and ViT-H as vision encoders. For each vision encoder size, we also vary patch size, and consider patch sizes of 32x32, 16x16, and 14x14, leading to $|M| = 15$ model configurations. We scale text encoders accordingly following previous literature [10]. For samples seen scales $D$, we consider a wide range of measurements $D = \{1.28\text{M}, 3.07\text{M}, 6.4\text{M}, 12.8\text{M}, 30.7\text{M}, 64\text{M}, 128\text{M}, 307\text{M}, 640\text{M}, 1.28\text{B}, 3.07\text{B}\}$, a total of $|D| = 11$ configurations. See App. Tab. 16 for full details about models and number of samples we used in our experiments. To rule out effects of training trial-to-trial variance on measurements and scaling law derivation, we estimate variance of downstream performance across training repetitions in the control experiment (App. Tab. 7).

**Learning rate schedule**. In our experiments, we consider both cosine and constant learning rate schedulers. For cosine learning rate schedule experiments, we performed a single run for each model-samples seen pair, while for constant learning rate schedule, we only need to train once for each model size.

**Optimization**. We perform a hyper-parameter sweep for batch size, learning rate and warmup for each training run to avoid suboptimal solutions. We have observed that it is important, especially in small sample seen scales, as large batch sizes usually used in CLIP training will result in small number of optimization steps, making optimization suboptimal. For training, we used AdamW [24] as an optimizer with a weight decay of $0.2$. To avoid unstable training and loss spikes with larger models (e.g., ViT-L, ViT-H) we followed [10, 25] and used $\beta_1 = 0.9$, $\beta_2 = 0.95$, gradient clip norm of 1, warmup and mixed precision with `bfloat16`. See more details in App. Tab. 15 and 14.

## 2.2 Downstream evaluation

**Zero-shot classification**. We evaluate the top-1 zero-shot accuracy on 35 classification tasks from the DataComp evaluation suite [19]. It includes ImageNet-1k [26], ImageNet robustness to distribution shift datasets [27, 28, 29, 30, 31], and additional 29 classification tasks covering multiple domains. We follow the evaluation protocol of [10], i.e. using the same prompts, class names, and code base [32]. The full list of datasets we used in evaluation with description is included in Appendix F.

**Zero-shot retrieval**. We evaluate models on MS-COCO[33] image and text retrieval Recall@5 metrics, following [10].

**Segmentation**. We fine-tune for semantic segmentation on ADE20K [34], following [35], using 31 epochs and a 1500-step warmup [36], with a consistent $224 \times 224$ input and $14 \times 14$ patch size.

## 2.3 Scaling law derivation and fitting procedure

We vary both model architecture size (number of parameters of both text and vision towers), number of samples seen and patch size. In general, the relationship between compute and performance follows a power law: $\mathcal{L} = aC^b$, where $C$ is compute in FLOPs [10, 37], with the set of points $\{C, \mathcal{L}(C)\}$ located on the Pareto frontier. Since we use error rate on ImageNet-1k zero-shot classification, we follow [37, 38] and take into account the saturation ($B_c$) that occurs at the small compute scales due to the nature of classification tasks - the probability of predicting a class even at zero-compute is determined by the frequency of this class in the test set. On the other hand, tasks like zero-shot image classification cannot be solved with 100% accuracy due to task- and learning method-intrinsic performance ceiling [38, 37]. Accordingly, we require that our scaling law functional form satisfies:

$$\mathcal{L}(C) > 0 \text{ (strictly positive)}, \quad \lim_{C \to \infty} \mathcal{L}(C) = E \text{ (irreducible error)}, \quad \frac{d\mathcal{L}}{dC} < 0 \text{ (monotonic decrease)}.$$

Given above criteria, we obtain the following functional form for the error that satisfies all three (subscript $C$ specifies compute dependency):

$$\mathcal{L}(C) = A_c \cdot (C + B_c)^{-\alpha_c} + E_c, \ \alpha_c > 0 \tag{1}$$

For each combination of compute scale $C$ and model architecture, we take a point with the minimal error rate. In previous works [39], points with minimal loss were found by binning FLOP values into 1500 FLOP logarithmically spaced intervals. We use the same approach to find points with minimal error. Therefore, we obtain a mapping from each combination of number of parameters $N$ and samples seen $D$ to the compute $C$ (in GFLOPs). Using measurements $\{(C_i, Y_i)\}$ where $C_i$ is compute budget (GFLOPs) and $Y_i$ is error performance, we fit the curve only on points with compute budget below a threshold $\{(C_i, Y_i), C_i < C_{\text{threshold}}\}$. To quantify the quality of our fit, we use mean squared error on the remaining (held-out) points: MSE $= \frac{1}{n} \sum_{i:C_i \geq C_{\text{threshold}}} (\mathcal{L}(C_i) - Y_i)^2$.

## 2.4 Confidence intervals estimation

We estimate 95% confidence intervals for the model parameters by propagating the uncertainty from the estimated parameters. We compute the Jacobian of the model, $J$, with respect to the parameters at the extrapolated points. We then estimate the variance of our predictions as $\sigma^2 = J^\top \text{Cov}(\hat{\theta}) J$. Confidence intervals are then given by $\hat{y} \pm t_{\alpha/2,\, n-p} \cdot \sigma$, where $t_{\alpha/2,\, n-p}$ is the critical value from the Student's $t$-distribution at $\alpha = 0.05$.

## 2.5 Data efficiency and optimal dataset size estimation

To quantify the data efficiency of CLIP and MaMMUT, we fit a scaling law analogous to Equation 1:

$$\mathcal{L}(D) = A_D \cdot (D + B_D)^{-\alpha_D} + E_D \tag{2}$$

Here, $\mathcal{L}(D)$ denotes the expected error rate as a function of the number of samples seen $D$. For each unique data budget $D$, we extract the corresponding minimal error points $\{D, \mathcal{L}(D)\}$ using the Skyline Operator algorithm [40], which selects non-dominated configurations based on error rate.

To estimate the dataset size that is optimal for a given compute budget, we follow the approach of [9] and fit a power-law relationship of the form: $D_{\text{opt}} = D_0 \cdot C^a$, where $D_{\text{opt}}$ minimizes $\mathcal{L}(C, D)$ for a given compute budget $C$. The optimal dataset sizes are obtained by identifying $(C, D)$ pairs that yield minimal error rate under fixed compute constraints.

# 3 Results

## 3.1 Scaling laws for model and dataset comparison

We fit function that has a form of Eq. 1 on the obtained experimental data using methods described in Section 2.3. To avoid the confound of data repetition [41] we limit the data used for our scaling law fits by selecting only models that were trained on up to 3B. In Tab. 5a estimated parameters for both models (CLIP and MaMMUT) as well as for both downstream tasks (IN1k classification

| Model Candidate | Samples Seen Candidate | GFLOPs | IN1k acc1 Predicted (95% CI) |
|---|---|---|---|
| ViT-L-14 | 12.8B | 2.14e+12 | 0.796 (0.788, 0.804) |
| ViT-L-14 | 15.5B | 2.59e+12 | 0.800 (0.791, 0.808) |
| mammut-ViT-L-14 | 10.6B | 2.14e+12 | 0.816 (0.811, 0.821) |
| mammut-ViT-L-14 | 12.8B | 2.59e+12 | 0.820 (0.815, 0.826) |

Table 1: Estimation of IN1k performance for CLIP and MaMMUT on unseen compute scales using our scaling laws fits. Additionally, for each compute scale, we provide possible models and samples seen (assuming unique samples) sizes.

and MS-COCO retrieval) can be found. **MaMMUT consistently exhibits better scaling behaviour** compared to CLIP. This is reflected in smaller error rates at equivalent compute budget at larger scales after crossing a compute threshold that is consistently found to be between $10^{10}$ and $10^{11}$ GFLOPS across various datasets and tasks (Fig. 1, 2,3). Note that on smaller scales in the lower performance range, CLIP consistently outperforms MaMMUT, which then consistently takes over CLIP at larger compute scales in the higher performance range. This indicates better efficiency and generalization as we increase compute. This trend holds across:

- Pre-training datasets: DataComp-1.4B (Fig. 1), Re-LAION-1.4B (Fig. 2) and DFN-1.4B (Fig. 3).
- Downstream tasks: ImageNet-1k zero-shot image classification (see Fig. 1 (a), Fig. 2 (a)) and MS-COCO image retrieval (Fig. 1 (b), Fig. 2 (b)), ADE20K semantic segmentation (Fig. 6).
- Learning rate scheduler: cosine (Fig. 1) and constant (Fig. 5).

We validate our fits by fitting the laws only up to certain compute budgets $C_{\text{threshold}}$ and then extrapolating to larger ones. We use these extrapolated points to compute MSE, which allows us to check on quality of the obtained fits, observing that adding more points to the fit reduces MSE on held-out points and also reduces uncertainty of predictions. The measured performance falls well within the prediction confidence interval (App. Tab. 12). Detailed versions of scaling laws plots for CLIP and MaMMUT can be found in Appendix B, more details on validating fit quality are in Appendix C. As further evidence for the validity of derived scaling laws, we observe same consistent scalability trends across datasets and on further important downstream tasks, for instance on DataComp eval suite (Fig. 14), ImageNet robustness (Fig. 15), or on segmentation after fine-tuning (Fig. 6), see Sec. F.

In Tab. 1 we provide predictions on DataComp-1.4B for both MaMMUT and CLIP for unseen compute budgets 2.14e+12 GFLOPs (corresponds to CLIP ViT-L-14 trained on 12.8B image-text pairs) and 2.59e+12 GFLOPs (corresponds to MaMMUT ViT-L-14 trained on 12.8B samples). We see that our predictions favor MaMMUT over CLIP. As a prediction test on larger scales further away, for CLIP ViT-L-14 trained on 12.8B samples of DataComp-1.4B our prediction for ImageNet-1k 0-shot accuracy (79.6%) is close to performance of CLIP ViT-L-14 trained on 12.8B samples reported in the original DataComp work [19] - 79.2%. Note that the measured performance IN1K zero-shot 79.2% in the DataComp original work [19] was done with heavy samples repetitions (12.8B on DataComp-1.4B is about 9x), while our prediction is done for unique or low repetition scenario, which also might explain tendency to a higher performance in the predictions than observed in experiments on 12.8B samples seen scale (Tab. 1).

As evident from Fig. 9 and 10, **DFN-1.4B consistently provides stronger scalability** compared to DataComp and Re-LAION, for both CLIP and MaMMUT architectures and for both zero-shot ImageNet-1k classification and MS-COCO retrieval. Despite lower compute used for dataset comparison and higher uncertainty for the trends resulting from fewer measurements for DFN, measured trends are clear and consistent, allowing thus to draw conclusions favoring DFN-1.4B over other datasets in the comparison.

## 3.2 Data efficiency and compute-optimal dataset size

Fig. 4 illustrates that MaMMUT exhibits **superior data efficiency** relative to CLIP. In Fig. 4 (a) we see that MaMMUT achieves better performance on ImageNet-1k zero-shot image classification as the number of training samples increases. In Fig. 4 (b) MaMMUT requires fewer training samples to achieve compute optimal performance on ImageNet-1k zero-shot classification. This indicates that

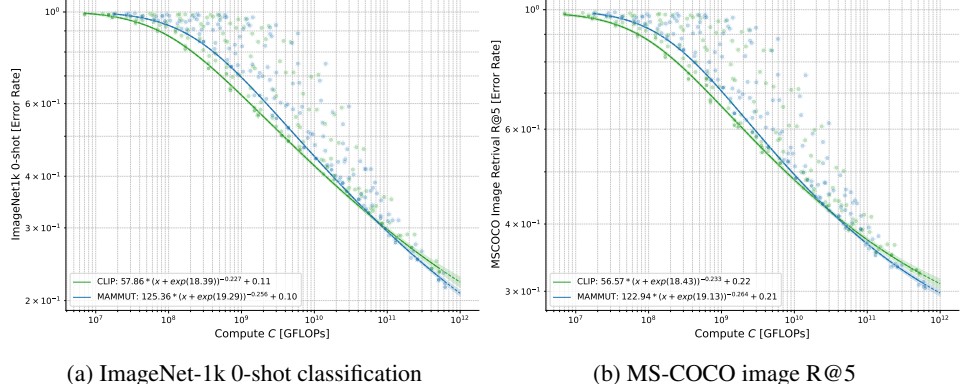

(a) ImageNet-1k 0-shot classification  (b) MS-COCO image R@5

Figure 1: **Scaling on DataComp-1.4B.** Comparison of CLIP and MaMMUT via scaling laws on DataComp-1.4B. Error rate on downstream tasks is plotted against compute. MaMMUT outperforms CLIP in terms of scalability, with scaling law fit lines crossing close to $10^{11}$ GFLOPS.

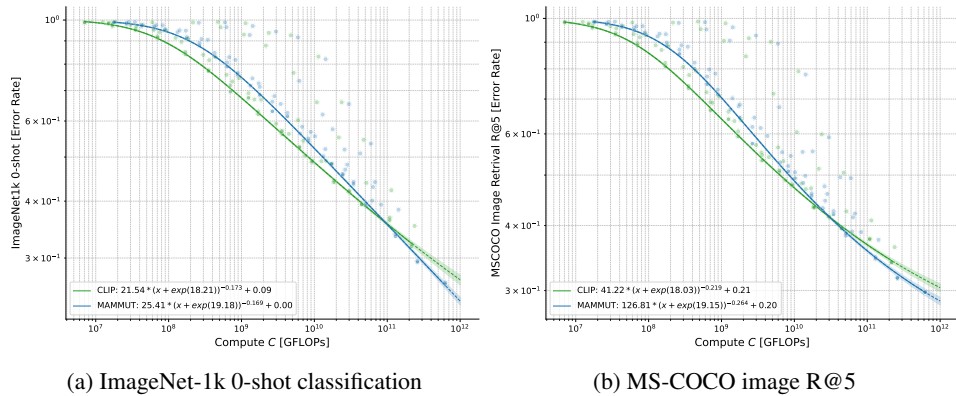

(a) ImageNet-1k 0-shot classification  (b) MS-COCO image R@5

Figure 2: **Scaling on Re-LAION-1.4B.** Comparison of CLIP and MaMMUT via scaling laws on Re-LAION-1.4B. Error rate on downstream tasks is plotted against compute. MaMMUT outperforms CLIP in terms of scalability, showing similar trends as in DataComp-1.4B.

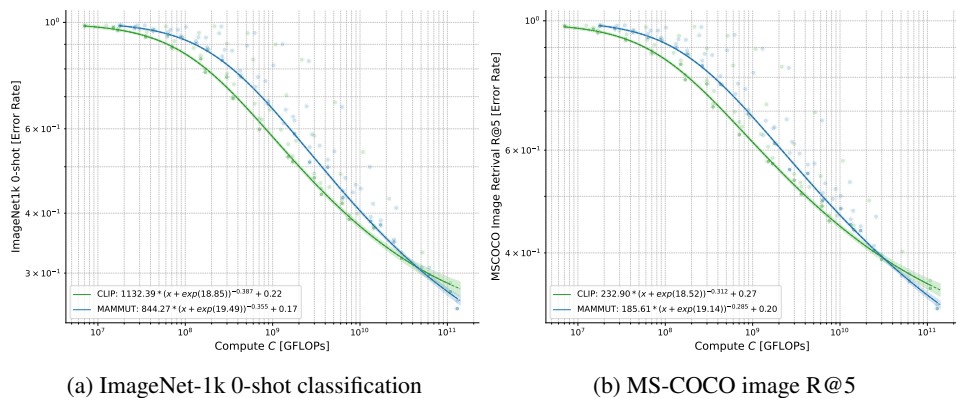

(a) ImageNet-1k 0-shot classification  (b) MS-COCO image R@5

Figure 3: **Scaling on DFN-1.4B.** Comparison of CLIP and MaMMUT via scaling laws on DFN-1.4B. Error rate on downstream tasks is plotted against compute. MaMMUT outperforms CLIP in terms of scalability, indicated by crossing scaling law fit lines, where MaMMUT takes over CLIP in performance from larger compute close to $10^{11}$ GFLOPS on, again showing similar trend as observed on DataComp and Re-LAION.

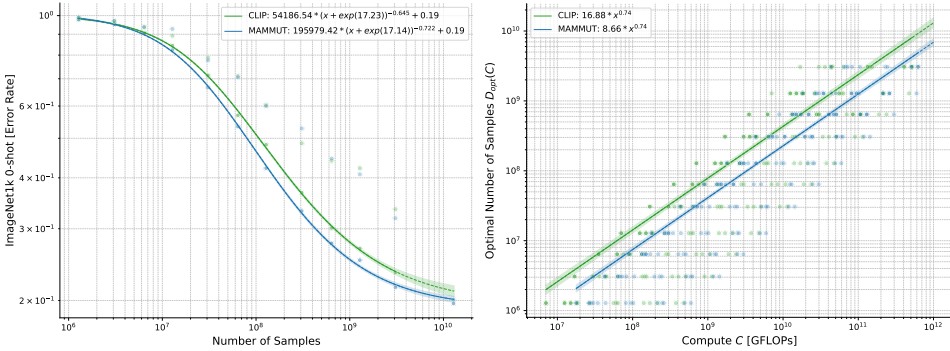

(a) IN-1k 0-Shot classification error rate vs. training dataset size

(b) Compute optimal dataset size $D_{opt}(C)$

Figure 4: Comparison of data efficiency and optimal dataset size for CLIP and MaMMUT via scaling laws on DataComp-1.4B. MaMMUT is more data efficient and requires smaller dataset size to be compute optimal.

MaMMUT makes **more effective use of training data** than CLIP. Therefore, these trends suggest that MaMMUT generalizes better and scales more favorably with additional data. Additionally, we derive scaling laws with respect to number of training samples on two other datasets - Re-LAION-1.4B and DFN-1.4B. In Fig. 11 we show that **MaMMUT shows consistently superior data efficiency across different training datasets** compared to CLIP. We also provide estimations for optimal number of training samples for unseen compute scales (see Tab. 2) for CLIP and MaMMUT trained on DataComp-1.4B. We use the predicted compute-optimal number of samples to estimate IN1k classification error rate using obtained power fit following Eq. 2 (see Fig. 4). We see that MaMMUT is a more scalable model which agrees with estimation obtained from fitting Eq. 1 on the experimental data (see Fig. 1 and Tab. 11).

|  | Predicted | Model |  | Predicted IN1k |
| --- | --- | --- | --- | --- |
| GFLOPs | $D_{opt}(C)$ (95% CI) | Candidate | #Params | 0-shot acc (95% CI) |
| 2.14e+12 | 2.30e+10 (2.75e+10, 1.91e+10) | ViT-SO150M-14-smaller-text | 279M | 0.794 (0.785, 0.803) |
| 2.59e+12 | 2.64e+10 (3.19e+10, 2.20e+10) | ViT-SO150M-14-smaller-text | 295M | 0.795 (0.786, 0.804) |
| 2.14e+12 | 1.23e+10 (1.39e+10, 1.09e+10) | mammut-ViT-L-14 | 522M | 0.798 (0.794, 0.803) |
| 2.59e+12 | 1.42e+10 (1.61e+10, 1.25e+10) | mammut-ViT-L-14 | 548M | 0.799 (0.795, 0.804) |

Table 2: Predicted compute optimal samples seen and accuracy for each compute budget and model configuration, with parameter sizes annotated in millions (M).

### 3.3 Scaling law derivation using constant learning rate scheduler

We follow [42] and show a scaling law derivation based on training with constant learning rate, thus saving 98% of compute compared to cosine. We omit points from warmup duration in our derivation, to prevent noise in the low-compute part of the scaling law. In Fig. 5 we visualize our results, showing the measurements density and in Tab. 6 we tabulate the coefficients. Our results further support the better scalability of MaMMUT over CLIP, showing consistent trends even when replacing learning rate scheduler.

### 3.4 Comparison via scaling law for fine-tuning error on segmentation dense prediction task

For further comparison evidence, we derive a scaling law (Eq. 1) for ADE20K segmentation error $(1 - \mathrm{mIoU})$ after fine-tuning dependent on pre-training compute scale for CLIP and MaMMUT. As shown in Fig. 6, MaMMUT again exhibits stronger scaling than CLIP ($\alpha = 0.208$ vs. $0.354$), with an error crossover at approximately $10^9$ GFLOPs. This is far below the crossover at approximately $10^{11}$ GFLOPs observed for zero-shot ImageNet classification (Fig. 1a), indicating that captioning supervision via fine-tuning improves dense prediction already at lower pre-training scales. See more details in Appendix H.

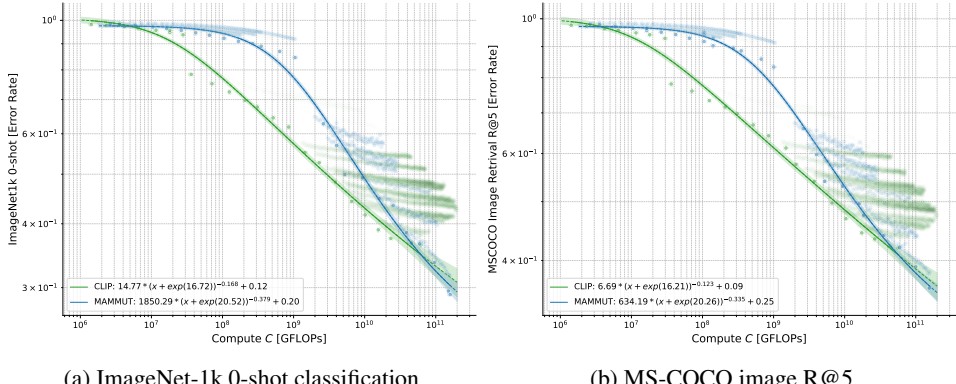

(a) ImageNet-1k 0-shot classification        (b) MS-COCO image R@5

Figure 5: **Scaling law fits using constant learning rate scheduler.** Comparison of CLIP and MaMMUT via scaling laws on DataComp-1.4B. Error rate on downstream tasks is plotted against compute. Using constant learning rate scheduler for scaling law derivation reveals the same trend as with cosine - MaMMUT outperforms CLIP in terms of scalability.

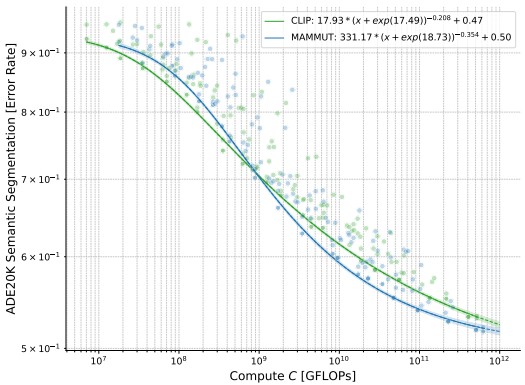

Figure 6: **Scaling law for semantic segmentation.** Downstream error rate $(1 - \mathrm{mIoU})$ of openCLIP and openMaMMUT pre-trained on DataComp-1.4B and fine-tuned on ADE20K. MaMMUT shows higher performance than CLIP for segmentation at higher scales.

### 3.5 Scaling up following the comparison: OpenMaMMUT L/14 on 12.8B samples seen scale

We used insights and predictions from our scaling analysis and comparisons to train OpenMaMMUT L/14 12.8B, a large scale open vision-language foundation model. We chose to train on 12.8B samples of DataComp-1.4B to have direct comparison to openCLIP trained in previous work [19]. More info on training hyperparameters can be found in Tab. 13. OpenMaMMUT achieves **state of the art performance** on zero-shot classification and retrieval tasks among similar-sized models trained only on publicly-available data (MetaCLIP, DataComp, OpenVision, Tab. 3). It outperforms with 80.3% IN1k accuracy as predicted openCLIP pre-trained on same DataComp-1.4B budget of 12.8B (79.2%) and even rivals models with much larger pre-training compute like SigLIP. OpenMaMMUT represents a highly performant, fully reproducible alternative to other models with **openly available data and training code**. Note that on 12.8B samples seen scale the performance suffers from high amount of repetitions, and therefore is below our prediction of 82% (Tab. 1) that is valid for training on unique samples. For further 12.8B sample scale models, see App. Sec. B.1

## 4 Related work & limitations

Recent work has investigated the performance of vision-language models such as CLIP [7], CoCa [16], MaMMUT [17], Cap [45], SigLIP [14], TULIP[46] or OpenVision[47] at various scales. These studies analyze different model sizes and highlight either architectural or dataset innovations;

| ViT | Res. | Seq. | Model | Dataset | #Samples | ImageNet-1k | | COCO | |
|---|---|---|---|---|---|---|---|---|---|
| | | | | | | val | v2 | T→I | I→T |
| L/16 | 256 | 256 | SigLIP [18] | WebLI-10B | 40B | 80.44 | 73.76 | 75.26 | 88.40 |
| | | | SigLIP 2 [14] | WebLI-10B | 40B | 82.35 | 76.66 | 76.84 | 90.44 |
| L/14 | 224 | 256 | OpenCLIP [10] | LAION-2B | 34B | 75.24 | 67.73 | 70.46 | 84.30 |
| | | | CLIP [7] | WIT-400M | 12.8B | 75.54 | 69.84 | 59.95 | 79.56 |
| | | | MetaCLIP [43] | MetaCLIP-2.5B | 12.8B | 79.19 | 72.64 | 71.36 | 84.94 |
| | | | EVA-CLIP [44] | Merged-2B | 4B* | 79.75* | 72.92* | 70.68 | 85.26 |
| | | | DFN [20] | DFN-2B | 13B | 81.41* | 74.58* | 73.19* | 86.20* |
| | | | DataComp [19] | DataComp-1.4B | 12.8B | 79.19 | 72.06 | 69.86 | 84.64 |
| | | | **OpenMaMMUT (Ours)** | DataComp-1.4B | 12.8B | 80.34 | 73.78 | 71.19 | 85.88 |

Table 3: Zero-shot classification (accuracy) and retrieval (R@5) results. DFN used ImageNet/MS-COCO-finetuned model for data filtering; EVA-CLIP was initialized from models pre-trained on ImageNet. We use **bold** for best overall results, gray for models involving ImageNet/MS-COCO data as training data in pipeline, and underlined for best results without ImageNet/MS-COCO involvement.

however, they do not perform a comprehensive scaling law analysis. Furthermore, the datasets used in these works—such as WIT for OpenAI CLIP and WebLI for SigLIP—are **closed**, severely limiting reproducibility and making it impossible to study which of algorithmic or dataset interventions claimed to have beneficial effect on learning used in those studies indeed lead to better model performance. In App. Fig. 12, we compare open implementations of CoCa and SigLIP to openCLIP on DataComp-1.4B, finding no significant difference of SigLIP to standard CLIP in model scalability.

In [48], authors investigate how various language model families compare in terms of their scaling behavior. While it offers valuable insights into architectural trends, it does not use for comparison a full scaling law framework in the sense of jointly modeling loss or downstream task performance as a function of compute and dataset size varied systematically across multiple scales. As accuracy of predictions derived from such trends was not measured, it makes it unclear whether observed trends for various language model architectures are to be trusted and whether comparison would remain valid across various scenarios, which we demonstrate to be the case for scaling law based comparison.

Preparing grounds for this work, [10] derived first reproducible scaling laws for openCLIP using LAION datasets and performed comparison between open LAION-400M/5B and closed WIT. The work used however only few samples seen scales, while also going up to scales that are prone to strongly diminishing performance due to heavy sample repetition (6x on 12.8B and 17x on 34B sample seen scales). This affects extrapolations of the scaling law and thus the validity of comparisons based on it. Interestingly, in our work using much denser scaling law derivation without strong repetitions for higher prediction accuracy, we can confirm the dataset comparison in [10] using Re-LAION-1.4B, observing same trends for WIT being better on zero-shot classification (App. Fig. 7) while worse on retrieval (App. Fig. 8), providing further evidence for robustness of scaling law based comparison. Another data-centric work responsible for composing open DataComp-1.4B dataset [19] used measurements on few selected scales to compare datasets and decide for benefit of various dataset interventions. No scaling laws were derived to back up the comparisons, leaving unclear whether the observed trends will propagate across larger scales than taken for the comparisons.

Several works explored the effects of data constraints on scaling laws. Notably, studies investigating the scaling law behavior under dataset repetitions for language models [38, 41] and for CLIP[49] reported that high repetition factors can lead to heavily diminished performance compared to the Pareto front of scaling with unique data samples. Our experiments carefully limit the repetition factor to less than 3×, minimizing such confounding effects, assuming unique samples or only little sample repetition for comparisons to be valid on Pareto front with strong performance scaling.

Building on these foundations, our work provides a unified and empirically grounded scaling law analysis of vision-language models trained on open datasets. We explicitly model performance as a function of data and compute [8, 9], and compare multiple architectures under fully controlled, consistent and reproducible settings. Unlike prior work, our approach enables rigorous comparison of both data and compute efficiency, ensuring the consistency of the comparison across scales and training conditions.

**Limitations.** While our work provides a comprehensive analysis for language-vision models such as CLIP and MaMMUT trained on large-scale open foundation datasets, it also has several limitations:

**1)** We mostly use zero-shot setting for model evaluation (for classification and retrieval tasks), using fine-tuning only for segmentation. Linear probing, fine-tuning [10] or vision instruction-tuning [12] of the pre-trained vision encoders can provide further insights into validity and consistency of scaling laws based comparison. **2)** While open datasets we use have substantial scale - 1.4B unique samples - this still limits the ability of derived scaling laws to extrapolate to higher scales, as the effect of sample repetition has to be considered when conducting measurements above 1.4B. To test comparison validity and scaling law predictions on larger reference scales like 12.8B, larger scale open datasets are required. **3)** In our study we only looked at standard contrastive loss or contrastive and captioning loss objectives and did not incorporate further "loss mixtures" such as masking or diffusion-based losses. We also did not derive scaling laws for specific architectural components such as optimal number of parameters in text or vision tower, or other important properties of training procedure like image input resolution and patch size, or context length in general. In its current form, scaling laws based comparison has high computational cost, prohibiting naive incorporation of many factors that influence scalability of the training procedure.

## 5 Discussion & conclusion

In this work, we show how systematic learning procedure comparison can be performed via scaling law derivation under fully controlled, reproducible training conditions when using open foundation models and open datasets. We take as example scenario openCLIP [7, 22, 10] and openMaMMUT based on [17], two important open language-vision models relying either on image-text contrastive only or contrastive and captioning loss, trained on three important open reference datasets, DataComp-1.4B [19], Re-LAION-1.4B [21] and DFN-1.4B [20]. We show that deriving scaling laws gives comparison of model and dataset based on their estimated scalability for wide scale spans and for various downstream tasks, aligned on same total pre-training compute. Such comparison can be validated by checking consistency of scaling trends in different scenarios. For instance, open-MaMMUT scalability is stronger than openCLIP both on zero-shot classification and retrieval, also showing advantage for a wide scale span on segmentation, and across all three open datasets. Also, inconsistencies are insightful - for instance, DataComp-1.4B shows stronger scalability for both openMaMMUT and openCLIP for zero-shot classification while being slightly weaker for retrieval. Thus, none of these two datasets is the most scalable candidate across all downstream tasks, and scaling advantage there is task dependent. DFN on the other hand is consistently better than other datasets across downstream tasks and for both openMaMMUT and openCLIP.

Comparison via scaling laws offers better protection against misleading conclusions derived from comparison of only few selected points, especially when done on small scales only. On smaller scales, openCLIP outperforms stronger scalable openMaMMUT that takes over on larger scales. Remarkably, we observe the compute scale threshold where openMaMMUT takes over openCLIP to be consistently settled between $10^{10}$ and $10^{11}$ GFLOPS across datasets, zero-shot downstream tasks and learning schedules. This gives further evidence for the robustness of scaling law based comparison. To properly estimate such crossings, it is crucial to perform dense measurements on smaller scales and use fitting routines that allow for accurate extrapolation to larger scales. Efficient derivation of accurate scaling laws [42, 11] to determine factors affecting scalability of the learning procedure is thus an important topic for future work.

In our study, we used open datasets with 1.4B samples. While this is sufficient to demonstrate usefulness of scaling law based comparison, more accurate predictions for training at larger scales on unique samples require larger datasets. Those are also required to train larger scale models with predicted strong capabilities, as too many repetitions on smaller datasets might lead to diminished performance [41, 49], which we see in openMaMMUT L-14 trained on 12.8B samples, staying with zero-shot IN1k 80.3% below the predicted 82% (Tab. 1). Deriving scaling law correction for diminishing performance due to data repetitions as well as increasing scale of open datasets are important directions for future work.

While we show that robust and reproducible comparison via scaling law derivation is possible, it relies crucially on the whole pipeline to be fully open - including dataset composition, training itself, and downstream evaluation. We hope that our work will encourage the creation of more open artefacts, especially open datasets as those are still scarce [50, 19, 20, 21], to enable collaborative and reproducible progress towards stronger scalable open foundation models guided by independently verifiable and systematic comparison.

## Acknowledgements

MN, TP, MC and JJ acknowledge funding by the Federal Ministry of Education and Research of Germany (BMBF) under grant no. 01IS24085C (OPENHAFM), under the grant 16HPC117K (MINERVA) and under the grant no. 01IS22094B (WestAI - AI Service Center West), as well as funding by EU Horizon under grant no. 101214398 (ELLIOT) and co-funding by EU from EuroHPC Joint Undertaking programm under grant no. 101182737 (MINERVA) and from Digital Europe Programme under grant no. 101195233 (openEuroLLM).

Giovanni Puccetti is fully funded by the project "Italian Strengthening of ESFRI RI RE-SILIENCE" (ITSERR) funded by the European Union under the NextGenerationEU funding scheme (CUP:B53C22001770006).

We gratefully acknowledge the Gauss Centre for Supercomputing e.V. for funding this work by providing computing time through the John von Neumann Institute for Computing (NIC) on the supercomputer JUWELS Booster at Jülich Supercomputing Centre (JSC), EuroHPC Joint Undertaking for computing time and storage on the EuroHPC supercomputer LEONARDO, hosted by CINECA (Italy) and the LEONARDO consortium through an EuroHPC Extreme Access grant EHPC-EXT-2023E02-068 and in part by Iscra B Project REWATCH (Project Number: HP10BL323R), storage resources on JUST granted and operated by JSC and supported by Helmholtz Data Federation (HDF), computing time granted by the JARA and JSC on the supercomputer JURECA at JSC, and computing time granted on prototype JEDI via JUREAP (JUPITER Early Access Programm) grant at JSC. LAION further acknowledges storage grant by HuggingFace that allows us to provide convenient access to the output of the open-source research to broad community via HF repository.

Further thanks go for support provided by supercomputing facilities and their teams, especially to Damian Alvarez and Mathis Bode from Juelich Supercomputer Center (JSC, Germany) and to Laura Morselli from CINECA (Italy).

We also would like to express gratitude to all the people who are working on making code, models and data publicly available, advancing community based research and making research more reproducible. Specifically, we would like to thank all the members of the LAION Discord server[2] community and Open-$\Psi$ (Open-Sci) Collective[3] for providing fruitful ground for scientific exchange and open-source development.

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

# Appendix: Scaling Laws for Comparison of Open Foundation Language-Vision Models and Datasets

## A    Estimated parameters for scaling law fits

To complement main results for scaling law based comparison of CLIP and MaMMUT (Sec. 3.1, Fig. 1, 2), we provide exact numbers of scaling law fits for both openCLIP and openMaMMUT measurements on zero shot IN1K classification and MS-COCO retrieval downstream tasks (Tab. 5a). Estimated values of exponents in power laws alone do not tell which models are more scalable, as we use here the functional form with additive terms for both irreducible error and non-zero random model performance (Eq. 1). Apart from plot visualization attesting Mammut stronger scalability than CLIP (Fig. 1, 2), scalability can be also compared via computing derivatives of the obtained fit in selected compute points. Derivatives that have larger absolute values stand for larger slope (bigger rate of decrease) and thus indicate stronger scalability. In Tab. 5b we show derivatives computed for scaling law fits, obtaining larger derivatives for openMaMMUT than for openCLIP, confirming again stronger scalability for MaMMUT over CLIP.

## B    More details on scaling law derivation experiments

**Compute budget and energy consumption for the experiments.** In Tab. 4, we provide overview over the GPU hours and energy spent for scaling law derivation experiments. We provide separate calculation for different learning rate schedule types (cosine, constant learning rate and constant learning rate + cooldown), for different datasets (Re-LAION-1.4B and DataComp-1.4B) and for different GPU types (A100 and H100). Large fraction of resources was spent for reference cosine schedule based scaling law derivation on DataComp-1.4B. We see that despite higher density of possible measurements, const based schedules use substantially less compute.

**Detailed versions of scaling law plots.** In the more detailed versions of scaling law plots (Fig. 16 and 17) we see the separate scaling curves for each model size (cooler colors indicate smaller models). The **bigger models require larger sample seen scale** to unfold their performance advantage, with the performance lagging behind smaller scale models on same smaller compute scale, where larger models suffer from sample seen scale bottleneck. On the other hand, **for the higher compute and samples seen scales, smaller models tend to saturate**, indicating a bottleneck in model number of parameters.

| LR Scheduler | GPU | Dataset | MWh | GPU Hours |
|---|---|---|---|---|
| **NVIDIA A100** | | | | |
| cosine | NVIDIA-A100 | DataComp-1.4B | 2.59e+05 | 1.03e+06 |
| const-cooldown | NVIDIA-A100 | DataComp-1.4B | 1.43e+05 | 5.72e+05 |
| const | NVIDIA-A100 | DataComp-1.4B | 9.30e+04 | 3.72e+05 |
| cosine | NVIDIA-A100 | Re-LAION-1.4B | 3.91e+04 | 1.56e+05 |
| const-cooldown | NVIDIA-A100 | Re-LAION-1.4B | 1.70e+04 | 6.79e+04 |
| const | NVIDIA-A100 | Re-LAION-1.4B | 4.61e+03 | 1.84e+04 |
| | | **A100 subtotal:** | **5.56e+05** | **2.22e+06** |
| **NVIDIA H100** | | | | |
| cosine | NVIDIA-H100 | DataComp-1.4B | 2.09e+04 | 2.98e+04 |
| cosine | NVIDIA-H100 | Re-LAION-1.4B | 1.06e+04 | 1.52e+04 |
| | | **H100 subtotal:** | **3.15e+04** | **4.50e+04** |
| | | **Total:** | **5.87e+05** | **2.27e+06** |

Table 4: Total GPU compute and energy consumption for scaling law derivation experiments.

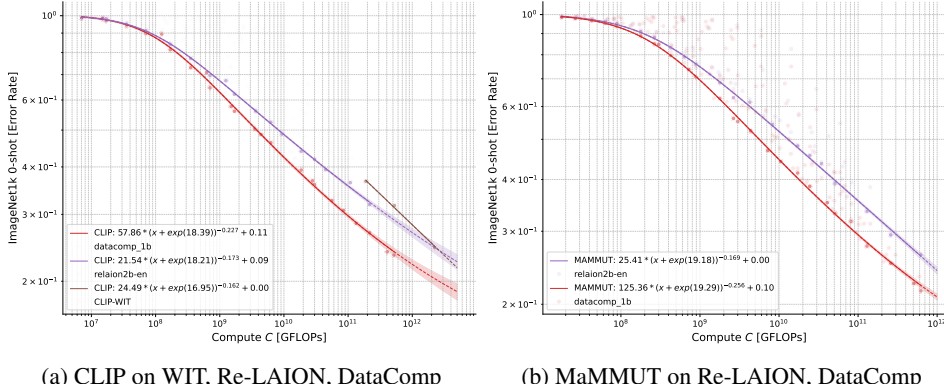

(a) CLIP on WIT, Re-LAION, DataComp   (b) MaMMUT on Re-LAION, DataComp

Figure 7: Scaling laws for IN1k 0-shot performance of openCLIP (left) and openMaMMUT (right), comparing training on DataComp-1.4B and Re-LAION-1.4B. For CLIP we have 3 additional points for OpenAI CLIP [7] models trained on WIT-400M dataset for reference.

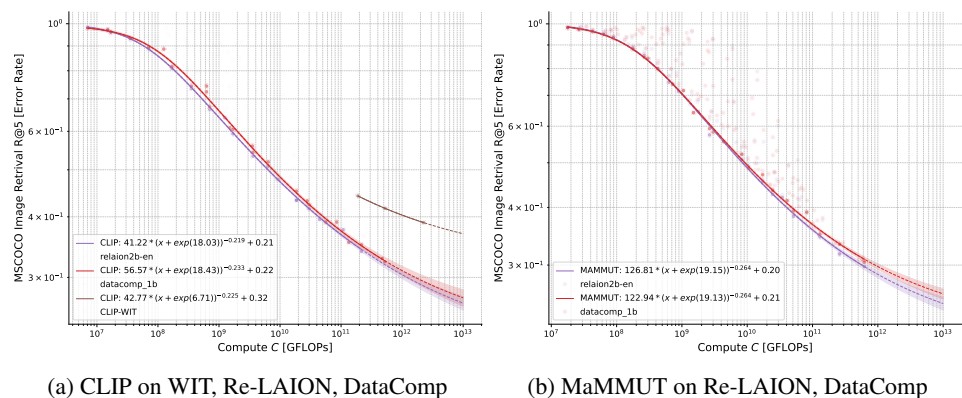

(a) CLIP on WIT, Re-LAION, DataComp   (b) MaMMUT on Re-LAION, DataComp

Figure 8: Scaling laws for MS-COCO image retrieval performance (1- Recall@5) of openCLIP (left) and openMaMMUT (right), comparing training on DataComp-1.4B and Re-LAION-1.4B. For CLIP models we have 3 additional points for OpenAI CLIP [7] trained on WIT-400M dataset for reference.

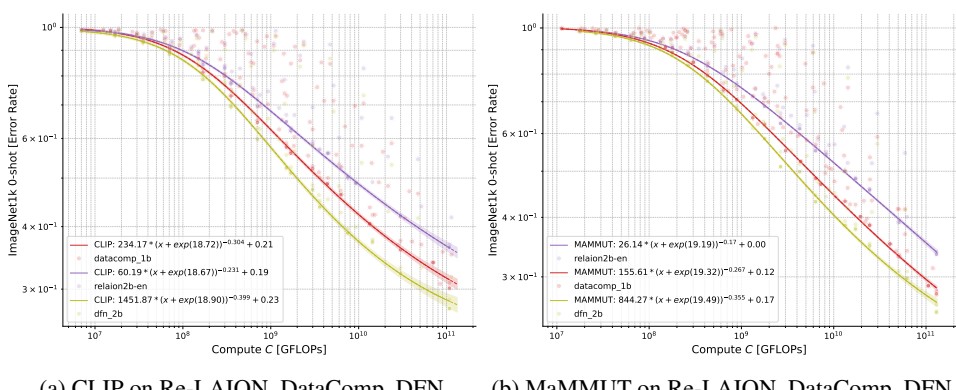

(a) CLIP on Re-LAION, DataComp, DFN   (b) MaMMUT on Re-LAION, DataComp, DFN

Figure 9: Scaling laws for IN1k 0-shot performance of openCLIP (left) and openMaMMUT (right), comparing training on Re-LAION-1.4B, DataComp-1.4B and DFN-1.4B. Training on DFN-1.4B results in superior performance across scales consistently for both architectures.

| Model | ImageNet-1k | | | | MS-COCO Retrieval | | | |
|---|---|---|---|---|---|---|---|---|
| | $A_c$ | $B_c$ | $\alpha_c$ | $E_c$ | $A_c$ | $B_c$ | $\alpha_c$ | $E_c$ |
| openCLIP | 57.862 | 18.391 | -0.227 | 0.111 | 53.913 | 18.413 | -0.230 | 0.216 |
| openMaMMUT | 79.970 | 19.111 | -0.233 | 0.076 | 119.751 | 19.122 | -0.263 | 0.212 |

(a) Fitted scaling law parameters $(A_c, B_c, \alpha_c, E_c)$ for error rate on 0-shot ImageNet-1k classification and MS-COCO retrieval tasks, rounded to three decimal places for models trained on DataComp-1.4B.

| $C_0$ GFLOPs | IN-1k Err. Rate $|d\mathcal{L}(C_0)/dC|$ | COCO R@5 Err. Rate $|d\mathcal{L}(C_0)/dC|$ |
|---|---|---|
| **CLIP** | | |
| 5.00e+10 | 9.85e-13 | 8.44e-13 |
| 1.00e+11 | 4.21e-13 | 3.60e-13 |
| 5.00e+11 | 5.86e-14 | 4.95e-14 |
| *Average: IN-1k: 4.882e-13, COCO: 4.177e-13* | | |
| **MaMMUT** | | |
| 5.00e+10 | 1.17e-12 | 9.65e-13 |
| 1.00e+11 | 4.92e-13 | 4.03e-13 |
| 5.00e+11 | 6.54e-14 | 5.28e-14 |
| *Average: IN-1k: 5.758e-13, COCO: 4.702e-13* | | |

(b) Numerical values of derivatives of fitted functions with respect to compute, in points $5 \cdot 10^{10}, 1 \cdot 10^{11}, 5 \cdot 10^{11}$ GFLOPs for both ImageNet-1k error rate and COCO retrieval error rate (1-R@5). MaMMUT consistently exhibits higher values of $|d\mathcal{L}(C_0)/dC|$ which corresponds to higher decrease rate and stronger scalability.

Table 5: Estimated parameters for main scaling law fits for 0-shot ImageNet-1k classification and MS-COCO retrieval, used for openCLIP and openMaMMUT comparison in Fig. 1

| Model | ImageNet-1k | | | | MS-COCO Retrieval | | | |
|---|---|---|---|---|---|---|---|---|
| | $A_c$ | $B_c$ | $\alpha_c$ | $E_c$ | $A_c$ | $B_c$ | $\alpha_c$ | $E_c$ |
| openCLIP | 14.769 | 16.725 | -0.168 | 0.121 | 6.686 | 16.209 | -0.123 | 0.089 |
| openMaMMUT | 1850.286 | 20.521 | -0.379 | 0.198 | 634.190 | 20.256 | -0.335 | 0.249 |

Table 6: Fitted scaling law parameters $(A_c, B_c, \alpha_c, E_c)$ for error rate on 0-shot ImageNet-1k classification and MS-COCO retrieval tasks, rounded to three decimal places for models trained on DataComp-1.4B with constant learning rate scheduler.

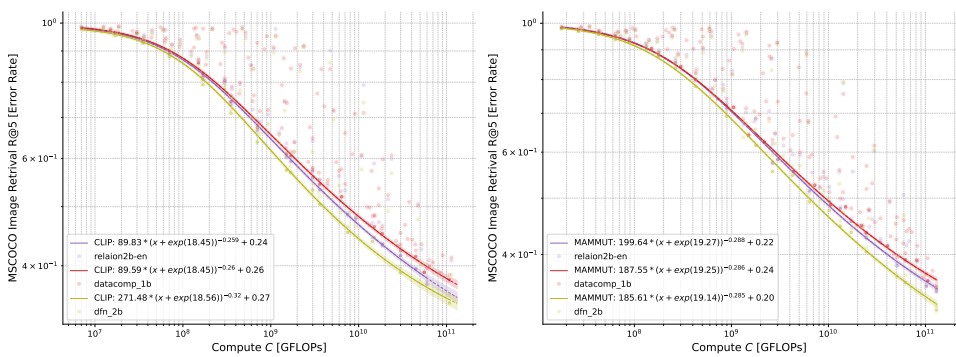

(a) CLIP on Re-LAION, DataComp, DFN   (b) MaMMUT on Re-LAION, DataComp, DFN

Figure 10: Scaling laws for MS-COCO image retrieval performance (1- Recall@5) of openCLIP (left) and openMaMMUT (right), comparing training on Re-LAION-1.4B, DataComp-1.4B and DFN-1.4B. Training on DFN-1.4B results again in superior performance across scales consistently for both architectures.

**Data efficiency on Re-LAION and DFN.** As we see from Fig. 11, MaMMUT exhibits consistently more superior scaling with respect on training data size on both Re-LAION-1.4B and DFN-1.4B. This supports the conclusion that **MaMMUT is more data efficient across multiple training datasets.**

**Training trial-to-trial variance.** To perform trial-to-trial variance sanity check for model pretraining, ensuring that trial-to-trial variance of same runs is substantially smaller that variance due to scaling or hyperparameter tuning, we show downstream task performance on zero-shot IN1K as

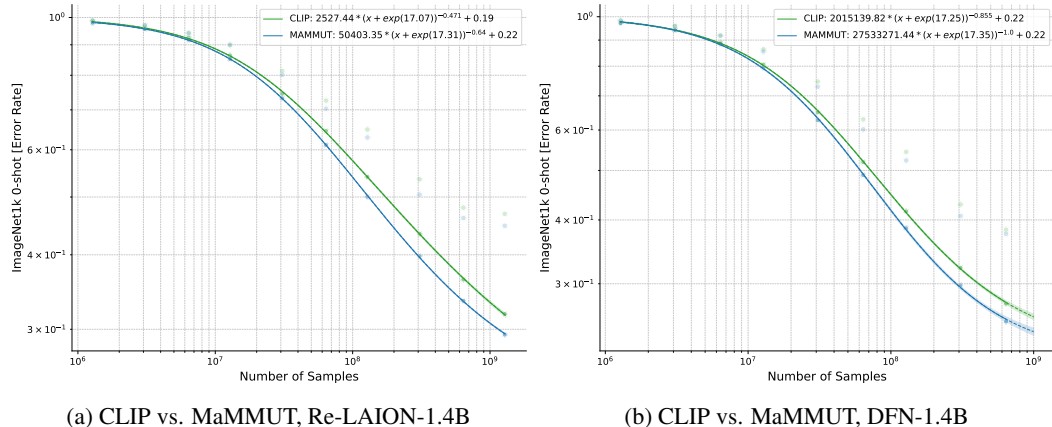

(a) CLIP vs. MaMMUT, Re-LAION-1.4B         (b) CLIP vs. MaMMUT, DFN-1.4B

Figure 11: Comparison of data efficiency for CLIP and MaMMUT via scaling laws for IN1k 0-shot classification error on Re-LAION-1.4B **(a)** and DFN-1.4B **(b)**. MaMMUT is consistently more data efficient on both datasets, which is also in accord with observations from DataComp-1.4B.

observed for 3 training runs of the same configuration for reference scales B-32 and B-16 on 640M samples, using same hyperparameters that correspond to minimum loss obtained in tuning, residing on Pareto front. As the results in Tab. 7 suggest, variance is negligibly small compared to difference due to changing the compute scale. This allows us to conclude that measurements we use for scaling laws derivation can be distorted only insignificantly by trial-to-trial training variance, and scaling trends we observe are valid and shaped dominantly by training compute.

| Trial/Training | 1 | 2 | 3 | $\mu \pm \sigma$ |
|---|---|---|---|---|
| B-32 640M | 0.58522 | 0.57866 | 0.58014 | $0.58134 \pm 0.0034407$ |
| B-16 640M | 0.66926 | 0.6668 | 0.6631 | $0.666387 \pm 0.00310073$ |
| L-14 640M | 0.72368 | 0.7203 | 0.72398 | $0.722653 \pm 0.00204356$ |

Table 7: Trial-to-trial variance control experiment. IN1k zero-shot top-1 on DataComp-1.4B, 640M samples seen. Mean $\mu$ and standard deviation $\sigma$ computed for each reference scale over 3 different training runs. Hyperparameters for each reference scale training run are fixed and correspond to hyperparameters tuned to obtain minimal loss via multiple sweeps for each given reference scale. Trial to trial variance is negligible small compared to performance difference across the scales and is decreasing with increasing performance level.

## B.1 Further models on 12.8B sample scale

We trained models of various scales B/32, B/16 on 12.8B sample scale on DataComp and DFN datasets, complementing results for L/14 from Sec. In Tab. 8 we show that predictions for 12.8B scale are consistently overestimating observed performance (for both B/32 and L/14 taken as examples). As already mentioned in Sec. 3.5, this is expected as predictions are done for unique samples seen, while observed performance comes from training with repetitions. We show further IN1-k 0-shot performance numbers for DataComp and DFN on 12.8B scale for openMaMMUT B/32 and B/16 in Tab. 9, in addition to openMaMMUT L/14 12.8B reported in Sec. 3.5. As predicted by scaling laws based comparison, DFN trained models consistently outperform DataComp on 12.8B scale.

## B.2 Effect of the number of points used for the scaling law fits

For scaling law derivation on DataComp-1.4B (Fig. 1), we used different number of points for MaMMUT (1010 points) and CLIP (672 points). Since MaMMUT architecture was never trained before on open data like DataComp, we had to perform more rigorous hyper-parameter search than for CLIP, hence we ended up with a larger set of measurements. To understand whether the number of measurements that we have obtained affects our conclusions we conduct additional experiment to

| Type | model | samples | gflops | acc1, w. rpt. | acc1 predicted, unique samples (CI 95%) |
|---|---|---|---|---|---|
| CLIP | B/32 | 12.8B | 1.89e+11 | 0.6864 | 0.699 (0.693, 0.706) |
| MaMMUT | B/32 | 12.8B | 3.50e+11 | 0.69468 | 0.710 (0.702, 0.718) |
| CLIP | L/14 | 12.8B | 2.14e+12 | 0.792 | 0.796 (0.788, 0.804) |
| MaMMUT | L/14 | 12.8B | 2.59e+12 | 0.8034 | 0.820 (0.815, 0.826) |

Table 8: Predictions for compute budget corresponding to 12.8B samples seen scales, ViT B/32 and L/14 model scales derived from scaling law measured on DataComp-1.4B. Shown predictions for IN1K 0-shot classification accuracy, assuming compute-optimal training and unique or low repetitions samples, compared to measured performance on 12.8B samples from DataComp-1.4B, which results in about 9x repetition factor. As it is known that repetitions may result in diminished performance, the measured performance is expectedly lower than predictions for unique samples. Discrepancy between predicted and measured performance can be also due to training with given scale combinations not residing on compute-optimal Pareto frontier.

| Dataset | Model | Samples | IN1K 0-shot Acc |
|---|---|---|---|
| | B/32 | 12.8B | 0.6947 |
| DataComp | B/16 | 12.8B | 0.7546 |
| | L/14 | 12.8B | 0.8034 |
| DFN | B/32 | 12.8B | 0.7213 |
| | B/16 | 12.8B | 0.7773 |

Table 9: ImageNet-1K zero-shot classification accuracy for openMaMMUT models trained on 12.8B samples from DataComp-1.4B and DFN-1.4B datasets. Following predictions from scaling laws, DFN trained models consistently outperform DataComp on larger 12.8B scale.

double check whether there is any difference in the obtained scaling law if working with same number of points for MaMMUT as for CLIP on DataComp-1.4B. We perform bootstrapping, sampling randomly 672 points from 1010 available points for MaMMUT, doing 10 trials, fitting scaling law for each trial and averaging the obtained scaling law coefficients. We observe that the obtained fit coefficients have no significant difference from scaling law obtained with 1000 points 10. Thus, the

| Model | $A_c$ | $B_c$ | $\alpha_c$ | $E_c$ |
|---|---|---|---|---|
| CLIP | 57.862083 | 18.391321 | -0.226604 | 0.111169 |
| MaMMUT (full points) | 125.356572 | 19.289384 | -0.255670 | 0.101112 |
| MaMMUT (same points num. as CLIP) | 125.267163 | 19.301461 | -0.255934 | 0.108208 |

Table 10: Comparison of different models with their corresponding parameters.

comparison on reduced points shows similar trends. Measurements are balanced for Re-LAION (750 vs 703 points), as well as for DFN (737 vs 732 points).

## C   Evaluating scaling law fit quality

To validate our scaling law fits, we use a threshold $C_{\text{threshold}}$ to up which we take the data for the fit. We compute RMSE for the held-out points to get a measure of how good each fit is. We compare two $C_{\text{threshold}}$ values (see Tab. 11 and Fig. 20 for DataComp-1.4B dataset). We see that both RMSE and uncertainty (the width of the confidence intervals) decreases as we take more the more pointsfor the fit.

We also compare different functional forms that can be used to fit the data: model with double saturation ($\mathcal{L}(C) = A_c \cdot (C + B_c)^{-\alpha_c} + E_c$) and without a term for irreducible error $E_c$:

$$\mathcal{L}(C) = A_c \cdot (C + B_c)^{-\alpha_c} \tag{3}$$

We choose first $C_{\text{threshold}} = 2.5 \cdot 10^{11}$ GFLOPs and the second $C_{\text{threshold}} = 5 \cdot 10^{11}$ GFLOPs. As we see from Tab. 11 and Tab. 12 for both values of $C_{\text{threshold}}$ double saturation form (Eq. 1) has consistently lower RMSE than the function without irreducible error. RMSE on held out points provides thus a way to select among various scaling law fits the candidate that provides better prediction accuracy for unseen scales, which in our case is the fit obtained via double saturation functional form (Eq. 1).

We see that the same trend of reducing confidence intervals and thus reducing uncertainty of the predictions when taking more points for the scaling law fit holds also for other tasks like MS-COCO image retrieval and other pre-training dataset Re-LAION-1.4B (see Fig. 20 and Fig. 13 for comparison between ImageNet-1k classification and MS-COCO retrieval and Figs. 18, 19 for Re-LAION-1.4B).

When comparing predictions with actually measured downstream task performance, we see that accuracy for the held-out points is high (Tab. 11). For instance, we measure for 3B samples seen scale on held-out points for openMaMMUT ViT L-14 zero-shot IN1K 0.784, with prediction 0.777 and 95% confidence interval (0.771, 0.783), and for openMaMMUT ViT H-14 0.795, with prediction 0.801 and 95% CI of (0.793, 0.809). Similar accuracy is observed for openCLIP, with actual measurements falling within predicted confidence intervals. **The derived scaling laws provide thus solid ground for comparison on unseen scales that have low amount of repetitions** (less than 3x in case of 3B samples seen scale when training on DataComp-1.4B or Re-LAION-1.4).

As already discussed in Sec. 3.5, the prediction for performance of MaMMUT L-14 on the larger 12.8B samples seen scale (Tab. 1, zero-shot IN1K 0.820, 95% CI (0.815, 0.826)) is therefore only made for low repetition scenario, and to validate it, dataset size larger than currently used 1.4B samples (which gives around 9x repetitions for 12.8B samples seen scale) is required. The measured 0.803 for openMaMMUT L-14 on 12.8B (Tab. 3) is thus expectedly below the prediction, as performance is diminished due to high amount of repetitions, in line with observations by previous works [41, 49].

| Model | Samples Seen | GFLOPs | IN1k 0-shot acc | Predicted IN1k 0-shot acc (95% CI) | Predicted (more points) IN1k 0-shot acc (95% CI) |
|---|---|---|---|---|---|
| **CLIP** | | | | | |
| ViT-L-16 | 3.07e+9 | 4.07e+11 | 0.761 | 0.747 (0.738, 0.755) | – |
| ViT-L-14 | 3.07e+9 | 5.18e+11 | 0.766 | 0.753 (0.744, 0.762) | 0.759 (0.751, 0.766) |
| ViT-H-14 | 3.07e+9 | 1.14e+12 | 0.784 | 0.773 (0.761, 0.784) | 0.779 (0.770, 0.789) |
| *RMSE: 1.26e-02* | *RMSE (more points): 5.90e-03* | | | | |
| **MaMMUT** | | | | | |
| mammut-ViT-L-14 | 1.28e+9 | 2.59e+11 | 0.749 | 0.743 (0.737, 0.748) | – |
| mammut-ViT-L-14 | 3.07e+9 | 6.22e+11 | 0.784 | 0.773 (0.765, 0.781) | 0.777 (0.771, 0.783) |
| mammut-ViT-H-14 | 3.07e+9 | 1.43e+12 | 0.798 | 0.797 (0.787, 0.807) | 0.801 (0.793, 0.809) |
| *RMSE: 7.57e-03* | *RMSE (more points): 7.57e-03* | | | | |

Table 11: Predicting held-out points on compute-optimal Pareto front based on scaling law derivation for the functional form with double saturation (Eq. 1). To check prediction accuracy when extrapolating beyond points taken for the fit, we predict starting from different compute threshold values of $C_{\text{threshold}}^{CLIP} = 4.07 \cdot 10^{11}$, $C_{\text{threshold}}^{MaMMUT} = 2.59 \cdot 10^{11}$. $C_{\text{threshold}}$ points themselves are predicted by taking smaller $C_{\text{cutoff}} = 2.5 \cdot 10^{11}$. The last column contains updated predictions made after taking additional data points up to $C_{\text{threshold}}$, showing predictions that extrapolate 2.4 and 5.5 compute factor beyond the fit for MaMMUT, and 1.3 and 2.8 for CLIP. Both confidence interval and RMSE decrease as we take more points. RMSE is consistently lower than RMSE measured for functional form without irreducible error (Tab. 12).

| Model | Samples Seen | GFLOPs | IN1k 0-shot acc | Predicted IN1k 0-shot acc (95% CI) | Predicted (more points) IN1k 0-shot acc (95% CI) |
|---|---|---|---|---|---|
| **CLIP** | | | | | |
| ViT-L-16 | 3.07e+9 | 4.07e+11 | 0.761 | 0.769 (0.764, 0.773) | – |
| ViT-L-14 | 3.07e+9 | 5.18e+11 | 0.766 | 0.778 (0.774, 0.783) | 0.777 (0.773, 0.782) |
| ViT-H-14 | 3.07e+9 | 1.14e+12 | 0.784 | 0.806 (0.802, 0.811) | 0.805 (0.801, 0.809) |
| *RMSE: 1.55e-02* | *RMSE (more points): 1.72e-02* | | | | |
| **MaMMUT** | | | | | |
| mammut-ViT-L-14 | 1.28e+9 | 2.59e+11 | 0.749 | 0.757 (0.754, 0.760) | – |
| mammut-ViT-L-14 | 3.07e+9 | 6.22e+11 | 0.784 | 0.795 (0.792, 0.798) | 0.794 (0.791, 0.796) |
| mammut-ViT-H-14 | 3.07e+9 | 1.43e+12 | 0.794 | 0.825 (0.822, 0.828) | 0.824 (0.822, 0.827) |
| *RMSE: 1.98e-02* | *RMSE (more points): 2.26e-02* | | | | |

Table 12: Predicting held-out points on compute-optimal Pareto front based on scaling law derivation for the functional form without irreducible error (Eq. 3). Comparing prediction quality to the functional form with double saturation (Tab. 11), using same values for $C_{\text{cutoff}}$ and $C_{\text{threshold}}$. The last column contains updated predictions made after taking additional data points up to $C_{\text{threshold}}$. Both confidence interval and RMSE decrease as we take more points. RMSE is consistently higher than RMSE measured for functional form with double saturation that includes irreducible error (Tab. 11).

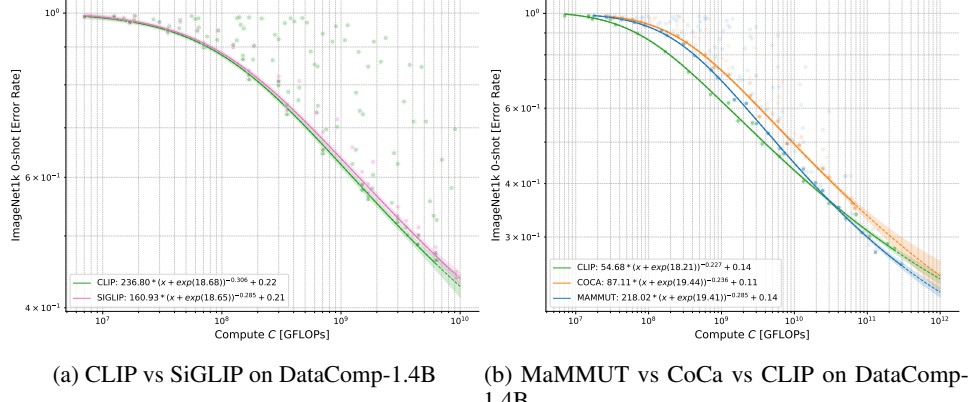

(a) CLIP vs SiGLIP on DataComp-1.4B

(b) MaMMUT vs CoCa vs CLIP on DataComp-1.4B

Figure 12: Scaling laws for ImageNet-1k 0-shot classification, comparing SigLIP (**left**) and CoCa (**right**) with standard CLIP and MaMMUT using open DataComp-1.4B dataset. SigLIP shows no benefit over standard CLIP, contrary to claims in previous work. CoCa is predicted to be less scalable than MaMMUT, while crossing CLIP is possible, although it is not clear due to high uncertainty for CoCa estimates on larger scales, as measurements on smaller scales for CoCa are not dense enough.

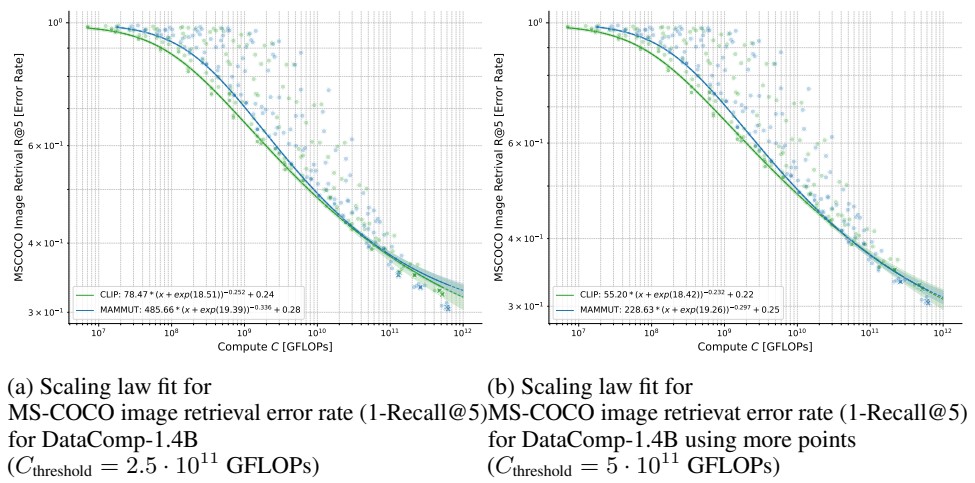

(a) Scaling law fit for
MS-COCO image retrieval error rate (1-Recall@5)
for DataComp-1.4B
($C_{\text{threshold}} = 2.5 \cdot 10^{11}$ GFLOPs)

(b) Scaling law fit for
MS-COCO image retrievat error rate (1-Recall@5)
for DataComp-1.4B using more points
($C_{\text{threshold}} = 5 \cdot 10^{11}$ GFLOPs)

Figure 13: Comparison of the fit quality for MS-COCO image retrieval error rate for openMaMMUT and openCLIP. Adding more points in (b) reduces the uncertainty of the fit, indicated by the width of bands around each curve.

## D  Datasets comparison

Scaling laws can be also be used as a tool for dataset comparison. Here, we compare performance of models trained on two reference datasets (DataComp-1.4B and Re-LAION-1.4B) for both model architectures – CLIP and MaMMUT, for two downstream tasks – ImageNet-1k 0-shot classification and MS-COCO retrieval. For CLIP, we additionally plot OpenAI CLIP models' performance that were trained on the WIT-400M dataset. As we see from Fig. 7, **for both CLIP and MaMMUT, training on DataComp-1.4B provides superior scalability for zero-shot ImageNet-1k classification**, compared to training on Re-LAION-1.4B. At the same time, training either on Re-LAION-1.4B or DataComp-1.4B leads to similar scalability and performance on MS-COCO retrieval, with Re-LAION-1.4B being for retrieval slightly more beneficial (Fig. 8).

Using much denser measurements for scaling law derivation, we can also confirm findings from previous work [10], which showed that the closed dataset WIT-400M[7] has better scaling trend on zero-shot classification, but worse scaling trend on zero-shot retrieval when compared to LAION-2B. We observe the same for Re-LAION-1.4B, which is a safety update of LAION-2B used in [10],

otherwise being same dataset with less samples due to link rot [21]. This provides further evidence for robustness of scaling law based comparison, showing consistent trends despite major difference in scaling law derivation. Previous work [10] used few samples seen scales of 3B, 12.8B and 34B, which results in high repetition given that only 2B unique samples are contained in LAION-2B [50], while our work used denser lower samples seen scales up to 3B, doing derivation with unique samples or low repetition only. Despite these differences, derived scaling laws agree in the dataset comparison for each downstream task, predicting same scaling trends in favor of WIT-400M on classification and in favor of Re-LAION-1.4B on retrieval. DataComp-1.4B can be seen in this comparison as an improved version of Re-LAION-1.4B, with stronger scalability on classification (Fig. 7) that matches WIT-400M, while obtaining performance for retrieval (Fig. 8) that matches Re-LAION-1.4B, outperforming WIT-400M.

# E    Scaling behavior of other architectures

We investigate additional model architectures: SigLIP [18] (CLIP with the sigmoid loss instead of softmax), CoCa [16] and Cap [45] (pure captioner). We train these models on DataComp-1.4B in order to compare with openCLIP and openMaMMUT. Fig. 12 shows the fitted lines for these models. We see that CLIP and SigLIP have very similar scaling behavior on ImageNet-1k classification (Fig. 12 (a)) while openMaMMUT consistently overtakes CoCa on the same compute scale Fig. 12 (b). Notably, our analysis shows that **SigLIP has similar or even worse scalability than CLIP** which contradicts recent claims of SigLIP being a better choice for a vision encoder [18, 14] due to its architectural advantages. Thus, when properly controlling for same training data in our experiments, no benefits for SigLIP can be derived from the obtained scaling law trends. We also observe that text **decoder-only MaMMUT overtakes encoder-decoder CoCa on the same compute scale**, indicating that simpler and more parameter efficient architecture of MaMMUT might be preferable.

Moreover, we see (Fig. 21) that MaMMUT has superior scaling compared to Cap, showing that **combination of contrastive and captioning losses is advantageous**. We see Cap also underperforming standard CLIP, hinting that Cap as captioner only based architecture is not a good candidate for strong scalability in 0-shot regimes, making another case for contrastive losses being important part of scalable architectures for 0-shot classification. It is further important to note that Cap can use only log-likelihood based evaluation for zero-shot classification task, as opposed to CLIP and MaMMUT that in addition can use embedding similarity based evaluation thanks to their contrastive loss. As evident from Fig. 21, embedding similarity based evaluation used in openCLIP and openMaMMUT has strong advantage over log-likelihood based one. It is in addition also much cheaper in execution. Cap has thus architectural disadvantage in not being able to use similarity based evaluation due to missing contrastive loss, which leads to inferior performance in 0-shot regime.

For both comparisons, we see uncertainty getting high when extrapolating to larger scales, which makes it for instance hard to predict whether CoCa might still cross CLIP or not. To reduce uncertainty, it is thus important to both conduct dense measurements at smaller scales and not to cut off measurements at scales too small to be used for proper extrapolation.

| Hyperparameter | Value |
|---|---|
| Model Architecture | mammut-ViT-L-14 |
| Samples Seen | 12.8B |
| Warmup Steps | 6000 |
| Global Batch Size | 180,224 |
| Learning Rate | $2.5 \times 10^{-3}$ |
| GPU Hours | $3.53 \times 10^4$ |
| Number of NVIDIA A100 GPUs | 1024 |

Table 13: Training hyperparameters for openMaMMUT-L-14.

| Model | Samples Seen | Warmup | Global Batch Size | Learning Rate |
|---|---|---|---|---|
| mammut-ViT-S-32 | 1.28e+06 | 1000 | 512 | 1.00e-03 |
| mammut-ViT-S-32 | 1.28e+06 | 1500 | 512 | 5.00e-04 |
| mammut-ViT-S-16 | 1.28e+06 | 1000 | 512 | 5.00e-04 |
| mammut-ViT-S-32 | 3.07e+06 | 4000 | 512 | 5.00e-04 |
| mammut-ViT-S-16 | 3.07e+06 | 4000 | 512 | 1.00e-03 |
| mammut-ViT-S-32 | 6.40e+06 | 4000 | 1024 | 1.00e-03 |
| mammut-ViT-S-32 | 1.28e+07 | 4000 | 2048 | 2.00e-03 |
| mammut-ViT-S-16 | 1.28e+07 | 3000 | 2048 | 2.00e-03 |
| mammut-ViT-S-32 | 3.07e+07 | 4000 | 4096 | 2.00e-03 |
| mammut-ViT-S-16 | 3.07e+07 | 3000 | 4096 | 2.00e-03 |
| mammut-ViT-S-32 | 6.40e+07 | 4000 | 4096 | 2.00e-03 |
| mammut-ViT-S-16 | 6.40e+07 | 4000 | 4096 | 1.50e-03 |
| mammut-ViT-S-32 | 1.28e+08 | 4000 | 8192 | 2.00e-03 |
| mammut-ViT-S-14 | 1.28e+08 | 4000 | 8192 | 2.00e-03 |
| mammut-ViT-M-16 | 1.28e+08 | 4000 | 8192 | 2.00e-03 |
| mammut-ViT-S-14 | 3.07e+08 | 4000 | 16384 | 2.00e-03 |
| mammut-ViT-M-16 | 3.07e+08 | 4000 | 16384 | 2.00e-03 |
| mammut-ViT-S-14 | 6.40e+08 | 4000 | 16384 | 1.50e-03 |
| mammut-ViT-B-16 | 3.07e+08 | 4000 | 16384 | 2.00e-03 |
| mammut-ViT-B-32 | 1.28e+09 | 4000 | 16384 | 2.00e-03 |
| mammut-ViT-B-16 | 6.40e+08 | 4000 | 32768 | 2.00e-03 |
| mammut-ViT-B-14 | 1.28e+09 | 4000 | 90624 | 2.00e-03 |
| mammut-ViT-L-16 | 6.40e+08 | 6000 | 45056 | 2.00e-03 |
| mammut-ViT-L-14 | 6.40e+08 | 6000 | 45056 | 2.00e-03 |
| mammut-ViT-L-14 | 1.28e+09 | 4000 | 90624 | 2.00e-03 |
| mammut-ViT-L-16 | 3.07e+09 | 4000 | 91136 | 2.00e-03 |
| mammut-ViT-L-14 | 3.07e+09 | 4000 | 91136 | 2.00e-03 |

Table 14: Hyperparameters for MaMMUT models trained on DataComp-1.4B that are located on the Pareto frontier

## F  Results on additional benchmarks

We also fit scaling laws on the data for other downstream tasks. In the Fig. 14 we show the scaling behavior on DataComp eval suite, which is constituted by averaging over 35 classification tasks from DataComp (see Tab.15 from [19]). Additionally, we provide scaling law fits for ImageNet-V2 and full ImageNet robustness set 0-shot classification performance for both openMaMMUT and openCLIP (Fig. 15). For all of these tasks we see the same trend - openMaMMUT is stronger scalable than openCLIP and has higher performance given the same compute at larger compute scales. This is also valid for the important robustness metrics that reflects out-of-distribution generalization (Fig. 15) - openMaMMUT shows stronger scalable robustness and outperforms openCLIP in robustness at larger compute scales.

## G  Additional training details

In Tab. 13 we provide hyperparameters that were used for training openMaMMUT L/14 at larger 12.8B scale. Additionally, in the Tab. 15 and 14 we provide training hyperparameters for all models and sample seen scales that were used for scaling law fits (i.e. models that are located on the Pareto frontier) for openMaMMUT and openCLIP respectively. In Tab. 16, we provide overview of hyperparameters used for the openCLIP and openMammut models that reside close to compute-optimal Pareto frontier. In Tab. 18, we show for selection of various model, sample seen scales and pre-training datasets hyperparameter combinations used for training for openCLIP and openMaMMUT. Tab. 17 shows hyperparameter ranges used for scaling law measurements for various architecture types. Finally, Tab. 19 contains samples of IN1K 0-shot accuracy numbers for the selection of model, sample seen scales and pre-training datasets for openCLIP and openMaMMUT.

| Model | Samples Seen | Warmup | Global Batch Size | Learning Rate |
|---|---|---|---|---|
| ViT-S-32 | 1.28e+06 | 1500 | 512 | 5.00e-04 |
| ViT-S-16 | 1.28e+06 | 1500 | 512 | 5.00e-04 |
| ViT-S-16 | 1.28e+06 | 1500 | 512 | 2.00e-03 |
| ViT-S-32 | 3.07e+06 | 1500 | 1024 | 5.00e-04 |
| ViT-S-32 | 6.40e+06 | 4000 | 1024 | 1.00e-03 |
| ViT-S-32 | 1.28e+07 | 4000 | 2048 | 1.00e-03 |
| ViT-M-32 | 1.28e+07 | 3000 | 2048 | 1.00e-03 |
| ViT-S-32 | 3.07e+07 | 4000 | 4096 | 2.00e-03 |
| ViT-S-32 | 6.40e+07 | 4000 | 4096 | 2.00e-03 |
| ViT-M-32 | 6.40e+07 | 10000 | 4096 | 1.00e-03 |
| ViT-S-32 | 1.28e+08 | 6000 | 8192 | 2.00e-03 |
| ViT-S-16 | 1.28e+08 | 6000 | 8192 | 2.00e-03 |
| ViT-S-32 | 3.07e+08 | 8000 | 16384 | 2.00e-03 |
| ViT-S-32 | 6.40e+08 | 4000 | 16384 | 2.00e-03 |
| ViT-S-14 | 3.07e+08 | 4000 | 16384 | 2.00e-03 |
| ViT-M-32 | 6.40e+08 | 6000 | 32800 | 2.00e-03 |
| ViT-B-32 | 1.28e+09 | 15000 | 16384 | 1.00e-03 |
| ViT-L-32 | 6.40e+08 | 4000 | 45056 | 2.00e-03 |
| ViT-B-16-text-plus | 6.40e+08 | 6000 | 32768 | 2.00e-03 |
| ViT-L-32 | 1.28e+09 | 4000 | 90624 | 4.00e-03 |
| ViT-L-16 | 6.40e+08 | 4000 | 45056 | 2.00e-03 |
| ViT-L-32 | 3.07e+09 | 4000 | 91136 | 4.00e-03 |
| ViT-L-14 | 1.28e+09 | 4000 | 90624 | 4.00e-03 |
| ViT-L-16 | 3.07e+09 | 4000 | 91136 | 4.00e-03 |
| ViT-L-14 | 3.07e+09 | 4000 | 91136 | 4.00e-03 |

Table 15: Hyperparameters for CLIP models trained on DataComp-1.4B that are located on the Pareto frontier.

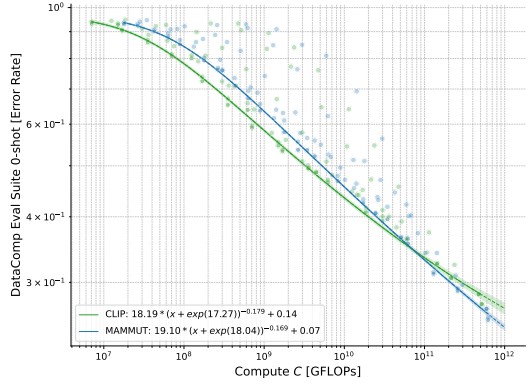

Figure 14: Scaling law on DataComp evaluation suite (average over 35 tasks, 0-shot classification), openCLIP vs. openMaMMUT comparison on DataComp-1.4B

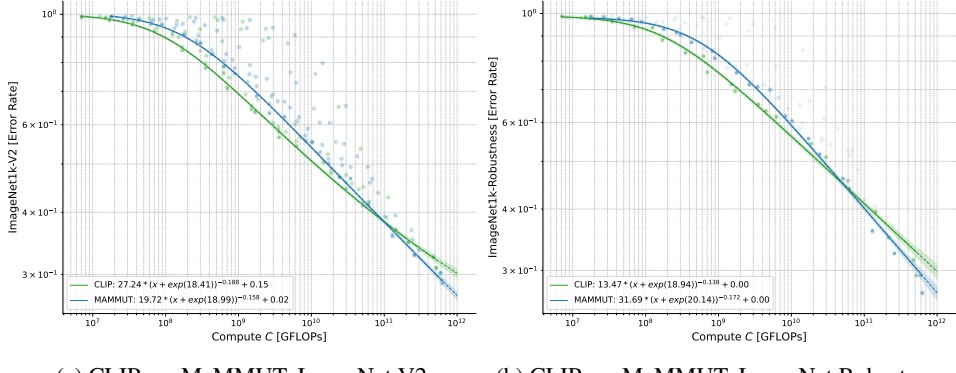

(a) CLIP vs. MaMMUT, ImageNet V2

(b) CLIP vs. MaMMUT, ImageNet Robustness

Figure 15: Scaling laws for ImageNet-v2 (left) and ImageNet robustness set (right, averaged performance across 5 datasets ImageNet-v2[27], ImageNet-R[28], ImageNet-Sketch[30], ObjectNet[31], and ImageNet-A[29]), 0-shot classification for openCLIP and openMaMMUT comparison on DataComp-1.4B

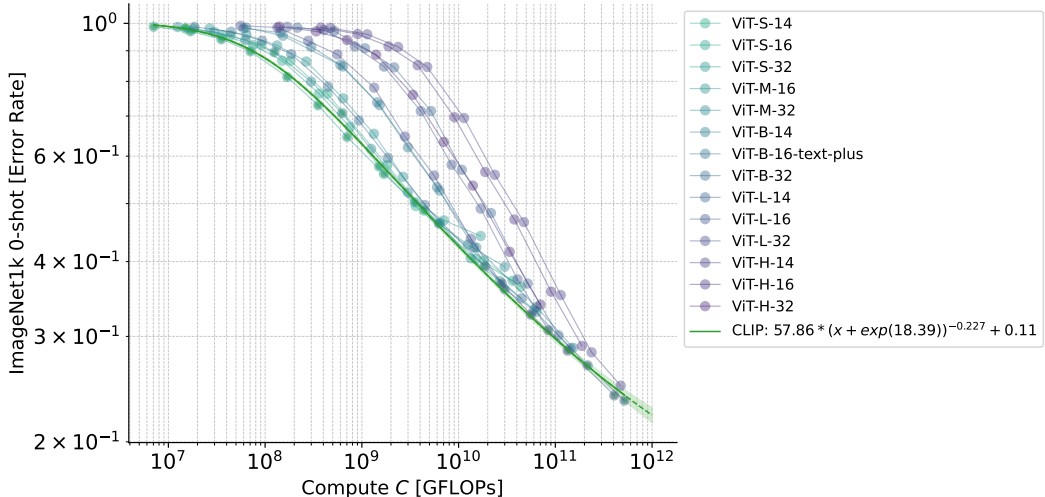

Figure 16: Detailed version of the scaling law fit for ImageNet 0-shot classification error rate for DataComp-1.4B for openCLIP. Cooler colors indicate smaller models. Bigger models are bottlenecked by samples seen scale (require larger samples seen than the smaller ones) and smaller models saturate with increased data and compute scale (over-training regime). Pareto front is composed by taking for each compute budget the points corresponding to models reaching minimal error rate for the given compute. Fit is performed through points on Pareto front.

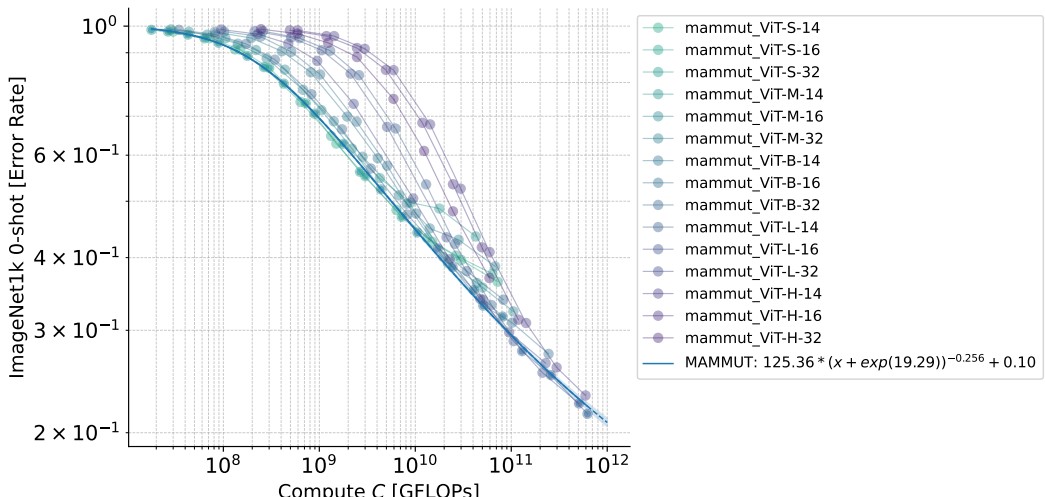

Figure 17: Detailed version of the scaling law fit for ImageNet 0-shot classification error rate for DataComp-1.4B for OpenMaMMUT. Cooler colors indicate smaller models. Bigger models are bottlenecked by samples seen scale (require larger samples seen than the smaller ones) and smaller models saturate with increased data and compute scale (overtraining regime). Pareto front is composed by taking for each compute budget the points corresponding to models reaching minimal error rate for the given compute. Fit is performed through points on Pareto front.

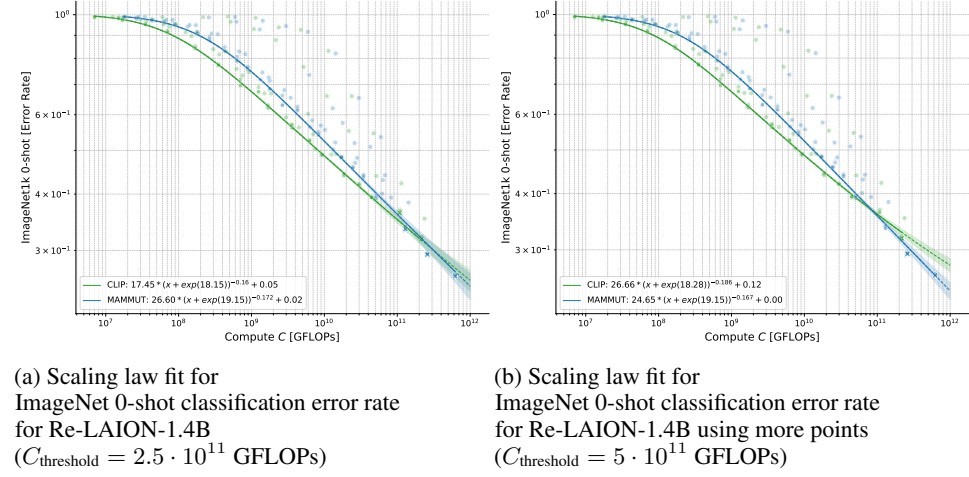

(a) Scaling law fit for
ImageNet 0-shot classification error rate
for Re-LAION-1.4B
($C_{\text{threshold}} = 2.5 \cdot 10^{11}$ GFLOPs)

(b) Scaling law fit for
ImageNet 0-shot classification error rate
for Re-LAION-1.4B using more points
($C_{\text{threshold}} = 5 \cdot 10^{11}$ GFLOPs)

Figure 18: Comparison of the fit quality for ImageNet-1k 0-shot classification error rate for openMaM-MUT and openCLIP trained on Re-LAION-1.4B. Adding more points in (b) reduces the uncertainty of the fit compared to (a), indicated by the width of bands around each curve.

| Name | Width | Emb | Depth | Params (M) | GFLOPs |
|---|---|---|---|---|---|
| ViT-S-32 | 384/384 | 384 | 12/12 | 63.09 | 5.51 |
| mammut-ViT-S-32 | 384/384 | 384 | 12/12 | 85.62 | 13.91 |
| ViT-S-16 | 384/384 | 384 | 12/12 | 62.26 | 11.75 |
| mammut-ViT-S-16 | 384/384 | 384 | 12/12 | 84.79 | 20.72 |
| ViT-S-14 | 384/384 | 384 | 12/12 | 62.21 | 14.3 |
| mammut-ViT-S-14 | 384/384 | 384 | 12/12 | 84.74 | 23.5 |
| ViT-M-32 | 512/512 | 512 | 12/12 | 103.12 | 9.74 |
| mammut-ViT-M-32 | 512/512 | 512 | 12/12 | 134.73 | 22.1 |
| ViT-M-16 | 512/512 | 512 | 12/12 | 102.02 | 20.84 |
| mammut-ViT-M-16 | 512/512 | 512 | 12/12 | 133.63 | 34.2 |
| ViT-M-14 | 512/512 | 512 | 12/12 | 101.95 | 25.37 |
| mammut-ViT-M-14 | 512/512 | 512 | 12/12 | 133.57 | 39.14 |
| ViT-B-32 | 768/512 | 512 | 12/12 | 151.28 | 14.54 |
| mammut-ViT-B-32 | 768/512 | 512 | 12/12 | 183.02 | 26.91 |
| ViT-B-16 | 768/512 | 512 | 12/12 | 149.62 | 39.51 |
| ViT-B-16-text-plus | 768/768 | 768 | 12/12 | 210.04 | 46.78 |
| mammut-ViT-B-16 | 768/512 | 512 | 12/12 | 290.52 | 79.7 |
| ViT-B-14 | 768/512 | 512 | 12/12 | 149.53 | 49.7 |
| mammut-ViT-B-14 | 768/512 | 512 | 12/12 | 181.27 | 63.54 |
| ViT-L-32 | 1024/768 | 768 | 24/12 | 429.95 | 43.59 |
| mammut-ViT-L-32 | 1024/768 | 768 | 24/12 | 510.63 | 74.28 |
| ViT-L-16 | 1024/768 | 768 | 24/12 | 427.74 | 132.37 |
| mammut-ViT-L-16 | 1024/768 | 768 | 24/12 | 508.42 | 165.37 |
| ViT-L-14 | 1024/768 | 768 | 24/12 | 427.62 | 168.61 |
| mammut-ViT-L-14 | 1024/768 | 768 | 24/12 | 508.29 | 202.56 |
| ViT-H-32 | 1280/1024 | 1024 | 32/24 | 989.02 | 109.81 |
| mammut-ViT-H-32 | 1280/1024 | 1024 | 32/24 | 1191.06 | 192.97 |
| ViT-H-16 | 1280/1024 | 1024 | 32/24 | 986.26 | 294.78 |
| mammut-ViT-H-16 | 1280/1024 | 1024 | 32/24 | 1188.3 | 385.72 |
| ViT-H-14 | 1280/1024 | 1024 | 32/24 | 986.11 | 370.28 |
| mammut-ViT-H-14 | 1280/1024 | 1024 | 32/24 | 1188.14 | 464.39 |

Table 16: Hyper-parameters of architectures we consider. **Width** refers to encoder width, **Emb** refers to embedding size, **Depth** refers to number of layers, **Params** refer to the number of parameters in millions, and **GFLOPs** refer to total GFLOPs per forward pass. Entries in the form of A / B denote image and text parameters respectively. There are more parameters in MaMMUT models because of the additional cross-attention layers.

## H   More details on fine-tuning for segmentation and scaling laws

Following prior work on how to benchmark vision foundation models for semantic segmentation [35], we evaluate CLIP and MaMMUT on semantic segmentation by fine-tuning them end-to-end using a linear decoder on ADE20K [34]. Regardless of the patch size used during pre-training, we interpolate the patch size of all models to $14 \times 14$, to ensure a fair comparison. We use an image input size of $224 \times 224$ and thus interpolate the positional embedding to $16 \times 16$. Hyperparameters used for training are consistent with [35], except the use of a linear learning rate warmup of 1500 steps, an epoch-based schedule of 31 epochs, and a batch size of 16 without gradient accumulation, following [36]. We fine-tune pre-trained models up to and including ViT-L and 3B samples seen, with different pre-training hyperparameters. We evaluate using a sliding window approach, again following [35].

Fig. 22 and Fig. 23 show the fitted scaling laws for CLIP and MaMMUT, respectively. Tab. 20 shows the corresponding estimated scaling law fit parameters.

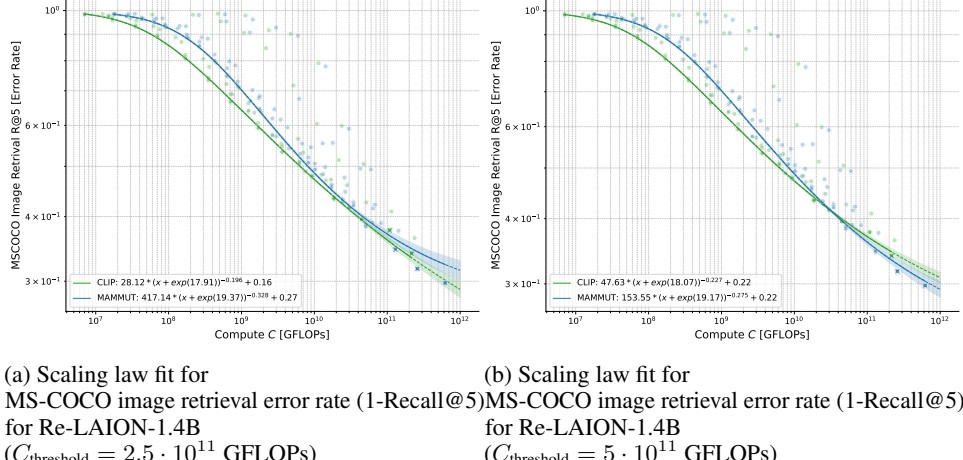

(a) Scaling law fit for
MS-COCO image retrieval error rate (1-Recall@5)
for Re-LAION-1.4B
($C_{\text{threshold}} = 2.5 \cdot 10^{11}$ GFLOPs)

(b) Scaling law fit for
MS-COCO image retrieval error rate (1-Recall@5)
for Re-LAION-1.4B
($C_{\text{threshold}} = 5 \cdot 10^{11}$ GFLOPs)

Figure 19: Comparison of the fit quality for MS-COCO image retrieval error rate for openMaMMUT and openCLIP trained on Re-LAION-1.4B. Adding more points in (b) reduces the uncertainty of the fit compared to (a), indicated by the width of bands around each curve.

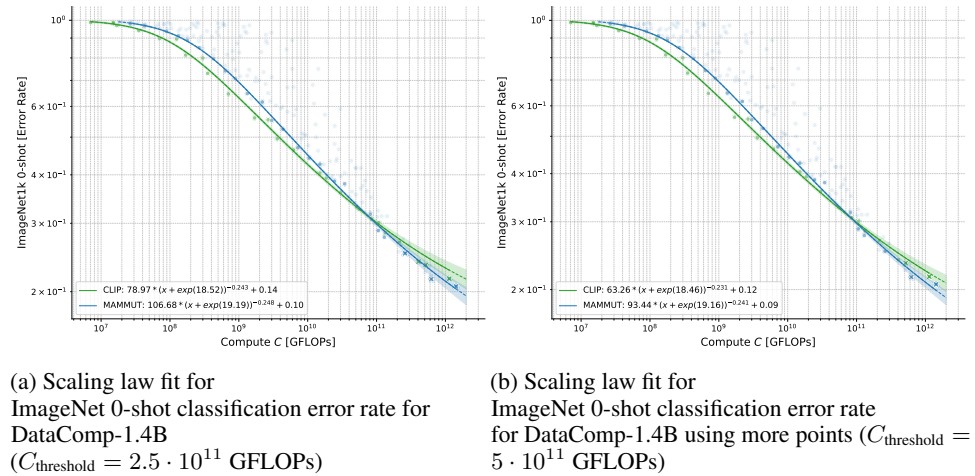

(a) Scaling law fit for
ImageNet 0-shot classification error rate for
DataComp-1.4B
($C_{\text{threshold}} = 2.5 \cdot 10^{11}$ GFLOPs)

(b) Scaling law fit for
ImageNet 0-shot classification error rate
for DataComp-1.4B using more points ($C_{\text{threshold}} = 5 \cdot 10^{11}$ GFLOPs)

Figure 20: Comparison of the fit quality for ImageNet-1k 0-shot classification error rate for openMaMMUT and openCLIP trained on DataComp-1.4B. Adding more points in (b) reduces the uncertainty of the fit compared to (a), indicated by the width of bands around each curve.

Table 17: Hyperparameters for different model architectures used for scaling law measurement experiments.

| Model architecture | Learning Rate | | Global Batch Size | | Warmup Steps | |
|---|---|---|---|---|---|---|
| | min | max | min | max | min | max |
| Cap | 1e-04 | 3e-03 | 512 | 90,624 | 1,000 | 20,000 |
| CLIP | 1e-04 | 1e-02 | 512 | 181,248 | 1,000 | 15,000 |
| CoCa | 1e-04 | 2e-03 | 512 | 45,056 | 1,500 | 10,000 |
| MaMMUT | 1e-05 | 1e-02 | 256 | 181,632 | 50 | 20,000 |
| SigLIP | 5e-05 | 2e-03 | 64 | 45,056 | 500 | 10,000 |

Table 18: Hyperparameters used for training to obtain scaling law measurements for openCLIP and openMaMMUT, shown for selection of different combinations of model, samples seen scales and pretraining datasets. NaN indicates that training for the particular combination was not performed.

| Dataset | Model Size | Samples Seen | Learning Rate | | Warmup | | Batch size | |
|---|---|---|---|---|---|---|---|---|
| | | | clip | mammut | clip | mammut | clip | mammut |
| DFN-1.4B | ViT-B-16 | 307M | 2.00e-03 | 1.50e-03 | 4.00e+03 | 4.00e+03 | 1.64e+04 | 1.64e+04 |
| DFN-1.4B | ViT-B-16 | 640M | 2.00e-03 | 2.00e-03 | 4.00e+03 | 6.00e+03 | 4.51e+04 | 3.28e+04 |
| DFN-1.4B | ViT-B-16 | 1B | 1.00e-03 | 1.50e-03 | 6.00e+03 | 6.00e+03 | 1.64e+04 | 3.28e+04 |
| DFN-1.4B | ViT-B-16 | 3B | NaN | 2.00e-03 | NaN | 6.00e+03 | NaN | 9.01e+04 |
| DFN-1.4B | ViT-B-32 | 307M | 1.50e-03 | 2.00e-03 | 4.00e+03 | 4.00e+03 | 1.64e+04 | 1.64e+04 |
| DFN-1.4B | ViT-B-32 | 640M | 1.00e-03 | 2.00e-03 | 6.00e+03 | 6.00e+03 | 1.64e+04 | 3.28e+04 |
| DFN-1.4B | ViT-B-32 | 1B | 2.00e-03 | 2.00e-03 | 6.00e+03 | 6.00e+03 | 9.01e+04 | 6.55e+04 |
| DFN-1.4B | ViT-B-32 | 3B | 2.00e-03 | 2.00e-03 | 1.00e+04 | 1.00e+04 | 6.55e+04 | 6.55e+04 |
| DFN-1.4B | ViT-H-14 | 307M | 1.50e-03 | 1.50e-03 | 4.00e+03 | 4.00e+03 | 3.28e+04 | 1.64e+04 |
| DFN-1.4B | ViT-H-14 | 640M | 2.00e-03 | 2.00e-03 | 4.00e+03 | 4.00e+03 | 4.51e+04 | 4.51e+04 |
| DFN-1.4B | ViT-L-14 | 307M | 2.00e-03 | 2.00e-03 | 4.00e+03 | 4.00e+03 | 3.28e+04 | 1.64e+04 |
| DFN-1.4B | ViT-L-14 | 640M | 2.00e-03 | 2.00e-03 | 6.00e+03 | 6.00e+03 | 3.28e+04 | 4.51e+04 |
| DFN-1.4B | ViT-L-14 | 1B | 4.00e-03 | 2.00e-03 | 4.00e+03 | 4.00e+03 | 9.01e+04 | 9.01e+04 |
| DataComp-1.4B | ViT-B-16 | 307M | 2.00e-03 | 2.00e-03 | 4.00e+03 | 4.00e+03 | 1.64e+04 | 1.64e+04 |
| DataComp-1.4B | ViT-B-16 | 640M | 2.00e-03 | 2.00e-03 | 6.00e+03 | 4.00e+03 | 3.28e+04 | 3.28e+04 |
| DataComp-1.4B | ViT-B-16 | 1B | 1.00e-03 | 1.00e-03 | 4.00e+03 | 4.00e+03 | 1.64e+04 | 1.64e+04 |
| DataComp-1.4B | ViT-B-16 | 3B | 2.00e-03 | 2.00e-03 | 4.00e+03 | 4.00e+03 | 4.53e+04 | 9.06e+04 |
| DataComp-1.4B | ViT-B-32 | 307M | 2.00e-03 | 2.00e-03 | 4.00e+03 | 4.00e+03 | 1.64e+04 | 1.64e+04 |
| DataComp-1.4B | ViT-B-32 | 640M | NaN | 1.50e-03 | NaN | 4.00e+03 | NaN | 1.64e+04 |
| DataComp-1.4B | ViT-B-32 | 1B | 1.00e-03 | 2.00e-03 | 1.50e+04 | 4.00e+03 | 1.64e+04 | 1.64e+04 |
| DataComp-1.4B | ViT-B-32 | 3B | 3.00e-03 | 2.00e-03 | 4.00e+03 | 4.00e+03 | 9.06e+04 | 4.53e+04 |
| DataComp-1.4B | ViT-H-14 | 307M | 2.00e-03 | 1.50e-03 | 6.00e+03 | 6.00e+03 | 1.63e+04 | 1.63e+04 |
| DataComp-1.4B | ViT-H-14 | 640M | 2.00e-03 | 2.00e-03 | 4.00e+03 | 6.00e+03 | 4.51e+04 | 3.20e+04 |
| DataComp-1.4B | ViT-H-14 | 1B | 2.00e-03 | 2.00e-03 | 6.00e+03 | 6.00e+03 | 9.01e+04 | 9.02e+04 |
| DataComp-1.4B | ViT-H-14 | 3B | 1.50e-03 | 1.50e-03 | 4.00e+03 | 6.00e+03 | 9.02e+04 | 9.02e+04 |
| DataComp-1.4B | ViT-L-14 | 307M | 1.50e-03 | 1.50e-03 | 3.00e+03 | 4.00e+03 | 3.28e+04 | 1.64e+04 |
| DataComp-1.4B | ViT-L-14 | 640M | 2.00e-03 | 2.00e-03 | 4.00e+03 | 6.00e+03 | 4.51e+04 | 4.51e+04 |
| DataComp-1.4B | ViT-L-14 | 1B | 4.00e-03 | 2.00e-03 | 4.00e+03 | 4.00e+03 | 9.06e+04 | 9.06e+04 |
| DataComp-1.4B | ViT-L-14 | 3B | 4.00e-03 | 2.00e-03 | 4.00e+03 | 4.00e+03 | 9.11e+04 | 9.11e+04 |
| Re-LAION-1.4B | ViT-B-16 | 307M | 1.50e-03 | 1.50e-03 | 4.00e+03 | 4.00e+03 | 1.64e+04 | 1.64e+04 |
| Re-LAION-1.4B | ViT-B-16 | 640M | 2.00e-03 | 2.00e-03 | 6.00e+03 | 4.00e+03 | 4.51e+04 | 4.51e+04 |
| Re-LAION-1.4B | ViT-B-16 | 1B | 2.00e-03 | 2.00e-03 | 6.00e+03 | 6.00e+03 | 6.55e+04 | 6.55e+04 |
| Re-LAION-1.4B | ViT-B-16 | 3B | 2.00e-03 | 2.00e-03 | 1.00e+04 | 6.00e+03 | 6.55e+04 | 9.01e+04 |
| Re-LAION-1.4B | ViT-B-32 | 307M | 2.00e-03 | 2.00e-03 | 4.00e+03 | 4.00e+03 | 1.64e+04 | 1.64e+04 |
| Re-LAION-1.4B | ViT-B-32 | 640M | 2.00e-03 | 2.00e-03 | 4.00e+03 | 6.00e+03 | 3.28e+04 | 3.28e+04 |
| Re-LAION-1.4B | ViT-B-32 | 1B | 2.00e-03 | 1.50e-03 | 6.00e+03 | 6.00e+03 | 6.55e+04 | 3.28e+04 |
| Re-LAION-1.4B | ViT-B-32 | 3B | 2.00e-03 | 2.00e-03 | 6.00e+03 | 6.00e+03 | 9.01e+04 | 9.01e+04 |
| Re-LAION-1.4B | ViT-H-14 | 307M | 2.00e-03 | 1.50e-03 | 4.00e+03 | 4.00e+03 | 3.28e+04 | 3.28e+04 |
| Re-LAION-1.4B | ViT-H-14 | 640M | 2.00e-03 | 1.50e-03 | 4.00e+03 | 4.00e+03 | 4.51e+04 | 4.51e+04 |
| Re-LAION-1.4B | ViT-H-14 | 1B | 2.00e-03 | 1.50e-03 | 6.00e+03 | 6.00e+03 | 9.01e+04 | 9.01e+04 |
| Re-LAION-1.4B | ViT-H-14 | 3B | NaN | 4.00e-03 | NaN | 6.00e+03 | NaN | 1.80e+05 |
| Re-LAION-1.4B | ViT-L-14 | 307M | 2.00e-03 | 2.00e-03 | 3.00e+03 | 3.00e+03 | 3.28e+04 | 3.28e+04 |
| Re-LAION-1.4B | ViT-L-14 | 640M | 2.00e-03 | 2.00e-03 | 4.00e+03 | 4.00e+03 | 4.51e+04 | 4.51e+04 |
| Re-LAION-1.4B | ViT-L-14 | 1B | 4.00e-03 | 2.00e-03 | 6.00e+03 | 4.00e+03 | 9.01e+04 | 9.01e+04 |
| Re-LAION-1.4B | ViT-L-14 | 3B | 2.00e-03 | 2.00e-03 | 4.00e+03 | 4.00e+03 | 9.01e+04 | 9.01e+04 |

Table 19: IN-1K 0-shot accuracy for selection of different scale combinations and pretraining datasets for openCLIP and openMaMMUT. NaN indicates that training for the particular combination was not performed.

| Dataset | Model Size | Total Samples Seen | clip | mammut |
|---|---|---|---|---|
| DFN-1.4B | ViT-B-16 | 307M | 0.6274 | 0.6516 |
| DFN-1.4B | ViT-B-16 | 640M | 0.6779 | 0.6991 |
| DFN-1.4B | ViT-B-16 | 1B | 0.7059 | 0.7268 |
| DFN-1.4B | ViT-B-16 | 3B | NaN | 0.7541 |
| DFN-1.4B | ViT-B-32 | 307M | 0.5524 | 0.5682 |
| DFN-1.4B | ViT-B-32 | 640M | 0.6104 | 0.6218 |
| DFN-1.4B | ViT-B-32 | 1B | 0.6402 | 0.6599 |
| DFN-1.4B | ViT-B-32 | 3B | 0.6943 | 0.6947 |
| DFN-1.4B | ViT-H-14 | 307M | 0.6955 | 0.7190 |
| DFN-1.4B | ViT-H-14 | 640M | 0.7477 | 0.7666 |
| DFN-1.4B | ViT-L-14 | 307M | 0.6777 | 0.7021 |
| DFN-1.4B | ViT-L-14 | 640M | 0.7258 | 0.7470 |
| DFN-1.4B | ViT-L-14 | 1B | 0.7609 | 0.7777 |
| DataComp-1.4B | ViT-B-16 | 307M | 0.5777 | 0.6148 |
| DataComp-1.4B | ViT-B-16 | 640M | 0.6405 | 0.6693 |
| DataComp-1.4B | ViT-B-16 | 1B | 0.6679 | 0.6908 |
| DataComp-1.4B | ViT-B-16 | 3B | 0.7133 | 0.7267 |
| DataComp-1.4B | ViT-B-32 | 307M | 0.5028 | 0.5257 |
| DataComp-1.4B | ViT-B-32 | 640M | NaN | 0.5852 |
| DataComp-1.4B | ViT-B-32 | 1B | 0.6076 | 0.6209 |
| DataComp-1.4B | ViT-B-32 | 3B | 0.6529 | 0.6613 |
| DataComp-1.4B | ViT-H-14 | 307M | 0.6483 | 0.6912 |
| DataComp-1.4B | ViT-H-14 | 640M | 0.7178 | 0.7410 |
| DataComp-1.4B | ViT-H-14 | 1B | 0.7517 | 0.7682 |
| DataComp-1.4B | ViT-H-14 | 3B | 0.7840 | 0.7986 |
| DataComp-1.4B | ViT-L-14 | 307M | 0.6323 | 0.6685 |
| DataComp-1.4B | ViT-L-14 | 640M | 0.6987 | 0.7240 |
| DataComp-1.4B | ViT-L-14 | 1B | 0.7317 | 0.7488 |
| DataComp-1.4B | ViT-L-14 | 3B | 0.7656 | 0.7845 |
| Re-LAION-1.4B | ViT-B-16 | 307M | 0.5104 | 0.5434 |
| Re-LAION-1.4B | ViT-B-16 | 640M | 0.5810 | 0.6094 |
| Re-LAION-1.4B | ViT-B-16 | 1B | 0.6257 | 0.6439 |
| Re-LAION-1.4B | ViT-B-16 | 3B | 0.6677 | 0.6830 |
| Re-LAION-1.4B | ViT-B-32 | 307M | 0.4401 | 0.4574 |
| Re-LAION-1.4B | ViT-B-32 | 640M | 0.5122 | 0.5181 |
| Re-LAION-1.4B | ViT-B-32 | 1B | 0.5601 | 0.5634 |
| Re-LAION-1.4B | ViT-B-32 | 3B | 0.6065 | 0.6075 |
| Re-LAION-1.4B | ViT-H-14 | 307M | 0.5898 | 0.6214 |
| Re-LAION-1.4B | ViT-H-14 | 640M | 0.6575 | 0.6843 |
| Re-LAION-1.4B | ViT-H-14 | 1B | 0.7029 | 0.7258 |
| Re-LAION-1.4B | ViT-H-14 | 3B | NaN | 0.7615 |
| Re-LAION-1.4B | ViT-L-14 | 307M | 0.5662 | 0.6020 |
| Re-LAION-1.4B | ViT-L-14 | 640M | 0.6365 | 0.6652 |
| Re-LAION-1.4B | ViT-L-14 | 1B | 0.6818 | 0.7061 |
| Re-LAION-1.4B | ViT-L-14 | 3B | 0.7206 | 0.7369 |

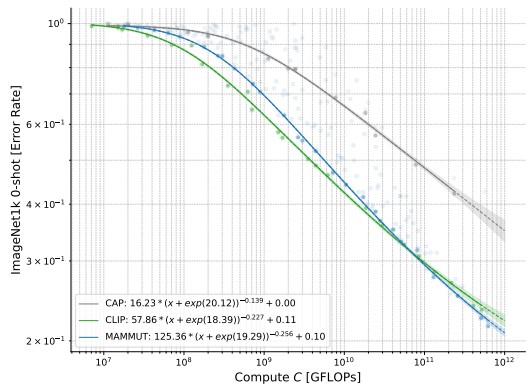

Figure 21: Scaling law fit for ImageNet-1k 0-shot classification, comparing MaMMUT, CLIP and Cap (captioning only). Cap can be only evaluated via log-likelihood, which is more expensive as similarity based evaluation used by CLIP and MaMMUT, as Cap misses contrastive loss in its architecture, which makes it disadvantageous for 0-shot setting.

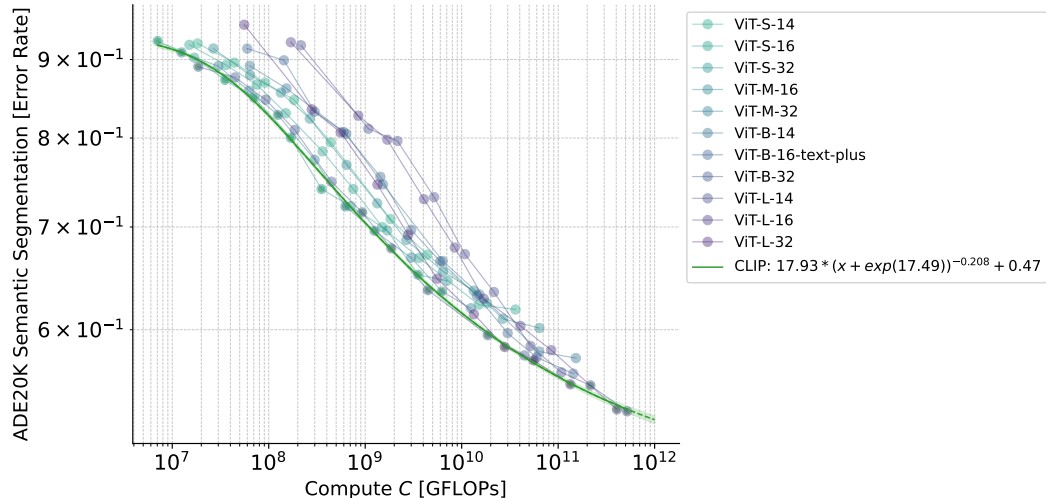

Figure 22: Downstream semantic segmentation performance of CLIP pre-trained on DataComp-1.4B and fine-tuned on ADE20K. Error rate $(1 - \mathrm{mIoU})$.

|  | $A_c$ | $B_c$ | $\alpha_c$ | $E_c$ |
|---|---|---|---|---|
| CLIP | 18.407549 | 17.577295 | -0.209187 | 0.468456 |
| MaMMUT | 352.152176 | 18.759619 | -0.356718 | 0.497617 |

Table 20: Fitted scaling law parameters $(A_c, B_c, \alpha_c, E_c)$ for segmentation error rate.

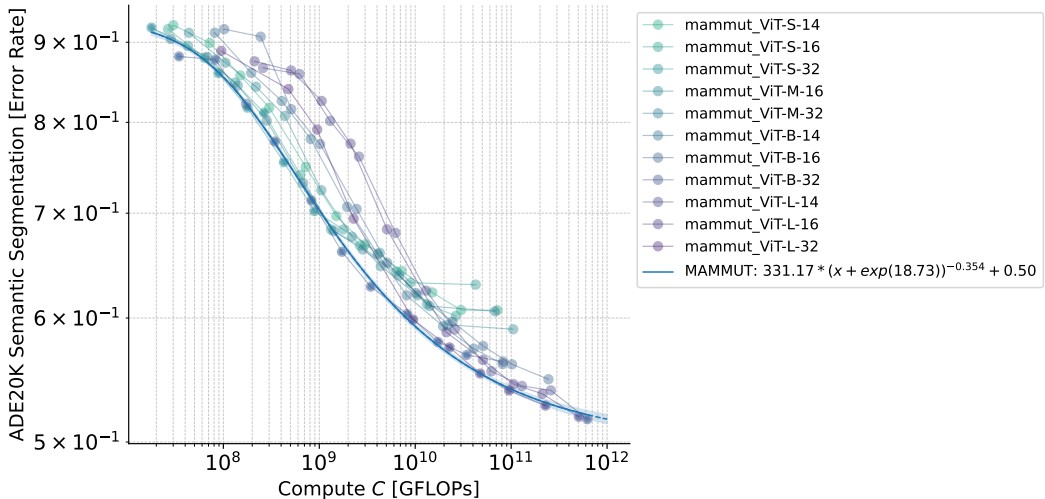

Figure 23: Downstream semantic segmentation performance of MaMMUT pre-trained on DataComp-1.4B and fine-tuned on ADE20K. Error rate $(1 - \text{mIoU})$.

# I   Author contributions

- **Marianna Nezhurina**: established major part of scaling law fitting procedures. Performed analysis of scaling law fit quality, derived predictions and confidence intervals. Conducted major part of data analysis. Performed initial training experiments with openCLIP and openMaMMUT. Established environments for experiments across various supercomputers. Supported compute resource acquisition. Established infrastructure for distributed dataset acquisition via Ray. Obtained Re-LAION and part of DFN dataset. Co-organized automated experiments data collection and analysis. Wrote the manuscript.

- **Tomer Porian**: co-designed and performed const lr schedule scaling law derivation experiments. Extended automated experiments execution for const lr schedule experiments. Collected and analyzed data, provided further input for scaling law fitting procedures. Co-wrote the manuscript.

- **Tommie Kerssies**: fine-tuning experiments for dense prediction segmentation, evaluating segmentation via different modes, scaling law derivation for segmentation, data collection and analysis. Co-wrote the manuscript.

- **Giovanni Puccetti**: openMaMMUT implementation in openCLIP, initial experiments with openCLIP and openMammut training. Initial co-design and implementation of automated experiments execution. Proof reading the manuscript.

- **Romain Beaumont** Re-LAION safety maintenance, hash filtering and re-packaging. Toolsets for dataset download and composition. Proof reading the manuscript.

- **Mehdi Cherti**: led the project, supported compute resource acquisition. Co-established environments for experiments across various supercomputers. Obtained DataComp, DFN and part of Re-LAION dataset. Designed and implemented automated experiments execution and evaluation. Wrote procedure for const lr schedule experiments. Conducted scaling law derivation experiments (DataComp, Re-LAION, DFN; openMammut, openCLIP, Cap). Designed and implemented evaluation. Organized automated experiments data collection and analysis. Collected and analysed the experimental data. Wrote the manuscript.

- **Jenia Jitsev**: led and coordinated the project, acquired compute resources. Organized data transfer (DataComp, Re-LAION) across the supercomputers. Co-established environments for experiments across various supercomputers. Co-designed automated experiments execution. Defined, designed and conducted scaling law derivation experiments (DataComp, Re-LAION, DFN; openMammut, openCLIP, CoCa, SigLIP). Collected and analysed the experimental data. Trained openMammut-L-14 on 12.8B of DataComp-1.4B, following the scaling law predictions. Led manuscript writing, wrote the manuscript.

