# OpenReview forum: "Scaling Laws for Robust Comparison of Open Foundation Language-Vision Models and Datasets"
_NeurIPS.cc/2025/Conference — NeurIPS 2025 poster_

### Official Review · Reviewer_Mi8q · 2025-06-19

**Clarity:** 3
**Significance:** 3
**Originality:** 2
**Rating:** 5
**Confidence:** 4

**Summary:**

This paper investigates scaling laws for CLIP and MaMMUT across multiple open datasets (Re-LAION-1.4B and DataComp-1.4B). Based on their experiments, the authors find that MaMMUT has both stronger improvement with scale and better sample efficiency than CLIP. They then derive scaling laws for downstream tasks such as classification, retrieval, and segmentation, and show that the observed improvements persist across these applications.

**Questions:**

1. The authors limit training to less than 3B samples to avoid repetition effects, but this constrains conclusions about truly large-scale behavior. When trained on 12.8B samples, performance falls below predictions. How can we be certain that this under-performance is actually due to high repetitions?

2. The paper states that SigLIP shows no benefits over CLIP (Section E), contradicting claims in the SigLIP papers. However, this comparison seems a bit limited. It also seems like training hyperparameters such as learning rate, batch size, etc. vary across scales. How were hyperparameters chosen to ensure fair comparison across architectures and scales? Was there a particular methodology for selecting these? How can we be confident that the observed scaling differences reflect architectural properties rather than differences in effort invested in optimization?

**Ethical Concerns:**

["NO or VERY MINOR ethics concerns only"]

**Final Justification:**

The authors have effectively addressed my concerns. The clarification on compute accounting resolves my questions about fair architectural comparison, while the extensive hyperparameter search and additional 12.8B sample experiments validate scaling law predictions and confirm persistent MaMMUT advantages.

While small concerns on having only zero-shot evaluation methods remain, I think these are acceptable given compute constraints and given justifications based on prior works.

Overall, this paper makes strong empirical contributions, and I think it will provide valuable open-source resources to the community. Thus, I am updating my rating to a 5.

**Limitations:**

Yes

**Quality:**

3

**Strengths And Weaknesses:**

Strengths:

The paper is clear and well-written. The claims are well supported both by theoretical scaling-law fitting and extensive experiments, with proper methodological rigour. There is great effort spent on ensuring a wide range of parameters are tested across various data regimes. The results will be open-sourced and reproducible, making it useful for the community at large. The release of openMaMMUT-L/14 will provide a good open alternative with performance competitive with proprietary vision-language models.

Weaknesses:

The scope of this paper is mostly focused on contrastive and captioning objectives – it would be interesting to see results on other alternative architectures and objectives. Evaluation primarily limited to zero-shot settings, with minimal analysis of few-shot learning, fine-tuning transfer, or vision-language understanding tasks. There is also some compute accounting ambiguity, especially when comparing architectures with inherently different parameter counts (Table 12), which raises questions about compute comparison methodology.

---

> ### Author Rebuttal · Authors · 2025-07-27
>
> We thank the reviewer for time dealing thoroughly with our work. We appreciate the reviewer’s remarks on the clarity and writing quality of the paper, highlighting the effort spent for methodologically rigorous and reproducible experiments, also mentioning usefulness for the open-source community. In following, we would like to address the relevant points raised by the reviewer:
>
> 1. > The scope of this paper is mostly focused on contrastive and captioning objectives - it would be interesting to see results on other alternative architectures and objectives.
>
> Available compute budget for the study allowed us to demonstrate execution of validated, reproducible comparison using scaling laws derivation for important architectures such as MaMMUT, CLIP, SigLIP, CoCa, CaP and for open datasets DataComp, Re-LAION and DFN (for DFN, we executed additional experiments for the rebuttal that will enter the revised manuscript). Comparison of further learning procedures requires additional compute budgets, and we hope that the introduced method will be used in follow up works to perform further useful, reproducible comparisons, guiding validated search for stronger scalable learning procedures.
>
> 2. > Evaluation primarily limited to zero-shot settings, with minimal analysis of few-shot learning, fine-tuning transfer, or vision-language understanding tasks
>
> Choosing zero shot evaluation for scaling laws (with exception of dense segmentation task where we performed fine-tuning) was due to two considerations. First is grounded in works like [1] and [2] stating that zero shot performance is a strong predictor for fine-tuning performance, and thus can be taken as good proxy for model quality. That is, stronger scalability for zero-shot would highly likely imply stronger scalability for full fine-tuning.  Second comes from available compute budget considerations: zero shot evals are cheap in compute, while full fine-tuning is expensive, especially given the large number of obtained points for scaling law derivation it has to be executed for. We thus think that zero-shot evals provide a strong proxy for model quality on the one hand, while allowing to keep compute costs down for evals on the other hand. In follow up studies, given further compute budgets, it will be interesting to perform fine-tuning evals for control as well.
>
> [1] Robust Fine-Tuning of Zero-Shot Models, Wortsman et al, CVPR 2022, arXiv:2109.01903 \
> [2] Reproducible Scaling Laws for Contrastive Language-Image Learning, Cherti et al, CVPR 2023, arXiv:2212.07143
>
> 3. > There is also some compute accounting ambiguity, especially when comparing architectures with inherently different parameter counts (Table 12), which raises questions about compute comparison methodology.
>
> We would like to clarify that there is no ambiguity wrt compute accounting. openCLIP profiler routine (relying on fvcore library) computes exact FLOPS for each model scale employed in our work. The calculated compute is then used for scaling law derivation and for reporting GFLOPS as listed in Table 12, taking into account image resolution, patch size, text context length and number of parameters in vision and text towers (compute does not depend on param count only!). This avoids any ambiguity, assigning for each model and samples scale its respective compute. The calculations can be reproduced and checked by installing openCLIP and calling profiler, eg ```python src/open_clip_train/profiler.py --model mammut_ViT-B-32 --results-file compute_profiling.csv``` for any given model config.
>
> 4. > … limit training to less than 3B samples to avoid repetition effects … constrains conclusions about truly large-scale behavior. When trained on 12.8B samples, performance falls below predictions. How can we be certain that this under-performance is actually due to high repetitions?
>
> Deriving scaling laws based on measurements up to 3.07B samples seen allows us to make predictions for behavior of compute-optimal training on unique or low repetition (2.2x for 1.4B) samples. We provide evidence that these predictions are accurate using held-out points on compute-optimal Pareto front for both CLIP and MaMMUT in App. Tab. 7, as we see actual measurements on compute-optimal Pareto front falling well into the 95% confidence interval for the predictions on held-out points. Predictions for MaMMUT extrapolate 2.4 and 5.5 compute factor beyond the fit (taking points up to $C^{MaMMUT}_{\text{threshold}} = 2.59\cdot 10^{11}$ GFLOPS for the fit). Factor 5.5 would allow us to extrapolate from 3B samples seen to 16.5B samples, which is well above 12.8B. As we confirm scaling law derivation giving accurate predictions when extrapolating far ahead, the most likely explanation for the deviation from predictions on 12.8B scale is thus sample repetition, as it is also known to cause diminishing returns from previous works [3], [4]. Further reason for observed performance drop can be also due to training procedure not residing on compute-optimal Pareto frontier. For rebuttal, we train in addition openMammut B/32 on 12.8B samples of DataComp-1.4B, resulting in following overview on 12.8B scale:
>
> | Type   | Model | Samples | GFLOPs    | acc1, w. rpt. | acc1 predicted, unique samples (CI 95%) |
> |--------|--------|---------|-----------|----------------|------------------------------------------|
> | CLIP   | B/32  | 12.8B   | 1.89e+11  | 0.6864         | 0.699 (0.693, 0.706)                     |
> | MaMMUT | B/32  | 12.8B   | 3.50e+11  | 0.69468        | 0.710 (0.702, 0.718)                     |
> | CLIP   | L/14  | 12.8B   | 2.14e+12  | 0.792          | 0.796 (0.788, 0.804)                     |
> | MaMMUT | L/14  | 12.8B   | 2.59e+12  | 0.8034         | 0.820 (0.815, 0.826)                     |
>
> For both B/32 and L/14, the measured IN1k 0-shot accuracy is below the predictions for unique samples, due to 9x sample repetitions (1.4B unique samples, 12.8B total), as elaborated above. The measured performance can be still projected from the predictions for unique samples by taking offset of 2\% for MaMMUT and 1\% for CLIP downwards from the confidence interval of the predicted performance on unique samples. Testing on unique samples is currently not possible, as it requires open datasets with scale of at least 6B unique samples to keep repetitions low. For accurate prediction on repeated samples, follow up work is required to derive scaling laws with corrections for sample repetitions which requires additional training experiments similar to [3], [4]. Important for our work is that comparison is valid also at larger scales, MaMMUT consistently outperforming CLIP as predicted from scaling laws, also for high sample repetition.
>
> [3] Scaling Data-Constrained Language Models. Muennighoff et al, NeurIPS 2023, arXiv:2305.16264 \
> [4] Scaling Laws for Data Filtering. Goyal et al, CVPR, 2024 arXiv:2404.07177
>
> 5. > … SigLIP shows no benefits over CLIP … How were hyperparameters chosen to ensure fair comparison across architectures and scales? … How can we be confident that the observed scaling differences reflect architectural properties rather than differences in effort invested in optimization?
>
> For our main comparison showcase, CLIP vs MaMMUT, wide hyperparams interval was scanned for both models extensively, resulting in 672 scans for CLIP and 1010 scans for MaMMUT on DataComp-1.4B. We invested more effort into MaMMUT tuning as CLIP architecture was already extensively tuned in previous works and we could afford to tune CLIP based on already known hyperparam settings, which is not the case of MaMMUT. MaMMUT as the previously less optimized architecture was thus tuned more to avoid unfair advantage for CLIP. Confirming there is no tuning advantage, predictions from scaling laws on held out points have same high accuracy and low uncertainty for both CLIP and MaMMUT, also showing similar RMSE (App. Tab. 7, as also elaborated above).
>
> Further, we conduct additional experiment to double check whether there is any difference in obtained scaling law if working with same number of points for MaMMUT as for CLIP on DataComp-1.4B. We perform bootstrapping, sampling randomly 672 points from 1010 available points for MaMMUT, doing 10 trials, fitting scaling law for each trial and averaging the obtained scaling law coefficients. We observe that the obtained fit coefficients have no significant difference from scaling law obtained with 1000 points:
>
> | Model                                           | Ac        | Bc        | αc        | Ec        |
> |------------------------------------------------|-----------|-----------|-----------|-----------|
> | CLIP                                            | 57.862083 | 18.391321 | -0.226604 | 0.111169  |
> | MaMMUT (full points)                            | 125.356572| 19.289384 | -0.255670 | 0.101112  |
> | MaMMUT (same points num. as CLIP, 10 random samples) | 125.267163| 19.301461 | -0.255934 | 0.108208  |
>
> Thus, comparison on reduced points shows again same trends. Measurements are balanced for Re-LAION (750 vs 703 points), as well as for DFN (737 vs 732 points). Comparison via scaling laws on Re-LAION (Fig. 2) and DFN gives same trends as on DataComp, showing MaMMUT stronger scalable than CLIP, again confirming no influence of tuning on comparison.
>
> For SigLIP and CLIP, we also have balanced tuning situation on DataComp-1.4B with 590 tuned points for SigLIP, and thus can rely on the scaling laws derived for the comparison (Fig. 8a). Executing controlled comparison on same open dataset DataComp 1.4B is sufficient to see that replacing softmax with sigmoid transfer function does not provide any benefit over CLIP. Original SigLIP study featured a closed dataset (WebLI), such that controlled comparison there is not possible. Our result shows that original SigLIP strong performance is highly likely due to the closed dataset WebLI, and not due to replacing softmax with sigmoid transfer function as claimed in the original work.

---

> ### Comment · Reviewer_Mi8q · 2025-08-05
>
> Thank you to the authors for the thorough response. My main concerns have been addressed.
>
> The extensive hyperparameter search and scope of experiments demonstrate appropriate rigor given the substantial compute investment. I am satisfied by the additional results in point 4 showing that the scaling laws remain valid and MaMMUT continues to outperform CLIP at larger scales. Additionally, the clarification regarding compute accounting using the openCLIP profiler methodology resolves my misunderstanding about fair comparison across architectures.
>
> I also think the SigLIP analysis suggesting that performance benefits may be dependent on WebLI, rather than architectural, is interesting and merits further investigation.
>
> Given the solid empirical contributions and valuable open-source resources this work will provide to the community, I will raise my score.

---

> ### Author Response · Authors · 2025-08-05
>
> We are delighted to hear our response was helpful to clarify the questions and thank the reviewer for engaging with timely reply and for raising the score.
>
> As reviewer is interested in the consistency of the comparison favoring MaMMUT vs CLIP across scales, we provide additional summary across selection of reference scales, showing actual measurements from the performed experiments for the 3 datasets (Re-LAION, DataComp, and DFN - addition for the rebuttal) on the Pareto front of best tuned models, with exact numbers for MaMMUT advantage (the "-" entries correspond to missing measurements, where also comparison is not possible)
>
> | Dataset           | Model Scale   | Samples Scale |  CLIP  | MaMMUT | Diff. (Mammut benefit) |
> |-------------------|--------------|-------------------:|-------:|-------:|------------------------:|
> | **Re-LAION-1.4B** | **ViT-B-32** | 307M               | 0.4401 | 0.4574 | +0.0173 |
> |                   |              | 640M               | 0.5122 | 0.5181 | +0.0059 |
> |                   |              | 1B                 | 0.5601 | 0.5634 | +0.0033 |
> |                   |              | 3B                 | 0.6065 | 0.6075 | +0.0010 |
> |                   | **ViT-B-16** | 307M               | 0.5104 | 0.5434 | +0.0330 |
> |                   |              | 640M               | 0.5810 | 0.6094 | +0.0284 |
> |                   |              | 1B                 | 0.6257 | 0.6439 | +0.0182 |
> |                   |              | 3B                 | 0.6677 | 0.6830 | +0.0153 |
> |                   | **ViT-L-14** | 307M               | 0.5662 | 0.6020 | +0.0358 |
> |                   |              | 640M               | 0.6365 | 0.6652 | +0.0287 |
> |                   |              | 1B                 | 0.6818 | 0.7061 | +0.0243 |
> |                   |              | 3B                 | 0.7206 | 0.7369 | +0.0163 |
> |                   | **ViT-H-14** | 307M               | 0.5898 | 0.6214 | +0.0316 |
> |                   |              | 640M               | 0.6575 | 0.6843 | +0.0268 |
> |                   |              | 1B                 | 0.7029 | 0.7258 | +0.0229 |
> |                   |              | 3B                 | -      | 0.7615 | -       |
>
>
> | Dataset       | Model Scale | Samples Scale |  CLIP  | MaMMUT | Diff. (Mammut benefit) |
> |---------------|------------|-------------------:|-------:|-------:|-----------:|
> | **DataComp-1.4B** | **ViT-B-32** | 307M | 0.5028 | 0.5257 | +0.0229 |
> |               |            | 640M | -    | 0.5852 | -     |
> |               |            | 1B   | 0.6076 | 0.6209 | +0.0133 |
> |               |            | 3B   | 0.6529 | 0.6613 | +0.0084 |
> |               | **ViT-B-16** | 307M | 0.5777 | 0.6148 | +0.0371 |
> |               |            | 640M | 0.6405 | 0.6693 | +0.0288 |
> |               |            | 1B   | 0.6679 | 0.6908 | +0.0229 |
> |               |            | 3B   | 0.7133 | 0.7267 | +0.0134 |
> |               | **ViT-L-14** | 307M | 0.6323 | 0.6685 | +0.0362 |
> |               |            | 640M | 0.6987 | 0.7240 | +0.0253 |
> |               |            | 1B   | 0.7317 | 0.7488 | +0.0171 |
> |               |            | 3B   | 0.7656 | 0.7845 | +0.0189 |
> |               | **ViT-H-14** | 307M | 0.6483 | 0.6912 | +0.0429 |
> |               |            | 640M | 0.7178 | 0.7410 | +0.0232 |
> |               |            | 1B   | 0.7517 | 0.7682 | +0.0165 |
> |               |            | 3B   | 0.7840 | 0.7986 | +0.0146 |
>
> | Dataset       | Model Scale   | Samples Scale |  CLIP  | MaMMUT | Diff. (Mammut benefit) |
> |---------------|--------------|-------------------:|-------:|-------:|------------------------:|
> | **DFN-1.4B**  | **ViT-B-32** | 307M               | 0.5524 | 0.5682 | +0.0158 |
> |               |              | 640M               | 0.6104 | 0.6218 | +0.0114 |
> |               |              | 1B                 | 0.6402 | 0.6599 | +0.0197 |
> |               |              | 3B                 | 0.6943 | 0.6947 | +0.0004 |
> |               | **ViT-B-16** | 307M               | 0.6274 | 0.6516 | +0.0242 |
> |               |              | 640M               | 0.6779 | 0.6991 | +0.0212 |
> |               |              | 1B                 | 0.7059 | 0.7268 | +0.0209 |
> |               |              | 3B                 | -      | 0.7541 | -       |
> |               | **ViT-L-14** | 307M               | 0.6777 | 0.7021 | +0.0244 |
> |               |              | 640M               | 0.7258 | 0.7470 | +0.0212 |
> |               |              | 1B                 | 0.7609 | 0.7777 | +0.0168 |
> |               | **ViT-H-14** | 307M               | 0.6955 | 0.7190 | +0.0235 |
> |               |              | 640M               | 0.7477 | 0.7666 | +0.0189 |
>
>
>
> We will update the manuscript accordingly with additional material and thank again for the discussion that was helpful to further clarify and improve the work.

---

### Official Review · Reviewer_jP4c · 2025-07-03

**Clarity:** 2
**Significance:** 4
**Originality:** 3
**Rating:** 5
**Confidence:** 3

**Summary:**

This paper has widely analyzed scaling laws for vision-language models (VLM), CLIP and MaMMUT, in terms of zero-shot classificaiton, retrieval, and segmentation. The trained VLMs on different massive datasets of DataComp and Re-LAION follow the novel scaling laws in different knowledge from the original paper of scaling laws. Finally, the analyzed VLMs perform well-established zero-shot classification accuracy after the training on a billion-scale dataset.

**Questions:**

1. Although the code release has been mentioned in the paper, what about the pre-trained models? The reviewer assumes these pre-trained models will also be made public. Having models and code available for image classification, image retrieval, and image segmentation would likely accelerate related researches across many subfields.

2. Are there effective pre-training approaches (e.g., parameters and how to pre-train) specialized for zero-shot classification, retrieval, and segmentation, respectively? Or does training on billion-scale datasets describe such specialization less important? If the underlying vision-language tasks share common characteristics, focusing on contrastive learning alone could suffice.

**Ethical Concerns:**

["NO or VERY MINOR ethics concerns only"]

**Final Justification:**

The reviewer is grateful to the authors for great efforts and the information. In particular, additional experimental results are very reasonable to further understand the paper and effectiveness. After the rebuttal phase, the reviewer has a better understanding on the research project. This has enough contributions to the topic on vision-language models. Hopefully the proposed method can be widely shared in the research community and employed diverse applications. Of course the reviewer keeps the accept-side rating in the final submission.

**Limitations:**

Yes

**Paper Formatting Concerns:**

No specific paper formatting concerns.

**Quality:**

4

**Strengths And Weaknesses:**

S1. The paper introduces a great effort to establish novel scaling laws across a couple of scenarios. While maintaining consistency with insights from previously reported scaling laws, the paper additionally covers new findings for the scaling laws in zero-shot image classification, image ritrieval, and image segmentation. The comprehensive analysis derives insights not only for zero-shot image classification but also for zero-shot image retrieval and image segmentation, presenting results that would be valuable to a broader AI community and researchers.

S2. The investigation is conducted with well-deserved strategy. In particular, it appears to have engaged in detailed discussion and try-and-error among themselves regarding the model architectures, datasets, training parameters, and learning strategies for arriving at their results. In terms of experimental design in VLMs, this paper serves as a useful knowledge for a wide readers.

S3. This paper includes excellent trial that the project itself is committed to "openness." Although the computational scale required to explore scaling laws is immense, making reproduction by many researchers challenging, the insights can be publicly and widely shared. Moreover, by open-sourced approaches, the community can directly benefit from the trial. The reviewer believes the paper's execution represents a significant contribution to the AI and realted fields.

S4. The paper shows that the dataset aspect is still tracing a larger scaling. At present, improving and organizing data can still accelerate VLM performance in classification, retrieval, and segmentation. In this point, this paper has the great potential to drive the overall advancement of the AI and related fields.

W1. The paper's constraints are understandable, however, the reviewer believes the paper’s value can be increased if more qualitative evaluations are included in the supplementary material as additional documents. In particular, since this paper uses OpenMaMMUT, which involves not only CLIP-style contrastive learning but also sentence generation, this would be beneficial to show examples of the kinds of captions generated for various images.

W2. Similar contributions such as Molmo & PixMo (CVPR 2025) aim to open-source VLMs. This work compares the accuracy against sophisticated models like GPT and Gemini at the time of paper submission, reporting comparable or slightly superior recognition performance. They also share many practical insights (e.g., introducing overlap among image patches), yielding interesting results. Although this work does not directly affect the contributions of the present paper, discussing their positioning and relationship would enhance paper's overall value.

---

> ### Author Rebuttal · Authors · 2025-07-28
>
> We thank the reviewer for time dealing thoroughly with our work. In following, we would like to address the relevant points raised by the reviewer:
>
> 1. > Although the code release has been mentioned in the paper, what about the pre-trained models? The reviewer assumes these pre-trained models will also be made public.
>
> This is correct. We will release the pretrained models (for the rebuttal, we additionally trained DataComp 12.8B openMammut B/32 which will complement  openMammut L/14 already described in the study) from all points measured for scaling law derivation (in total over 4500 models trained on DataComp, Re-LAION and DFN [1], for which we conducted further scaling law derivation experiments for the rebuttal), including intermediate training checkpoints for each model, their evaluation and source code to reproduce training, evaluation  and fitting of scaling laws. In the table below is the overview of all points measured during the study (including experiments on DFN [1] performed for the rebuttal) for which models will be released.
> | Dataset   | Model | Points |
> |-------------------|-------------|--------|
> | **DataComp-1.4B** | CLIP        | 672    |
> |                   | MaMMUT      | 1010   |
> | **Re-LAION-1.4B** | CLIP        | 703    |
> |                   | MaMMUT      | 750    |
> | **DFN-1.4B**        | CLIP        | 732    |
> |                   | MaMMUT      | 737    |
>
> [1] Data Filtering Networks. Fang et al, ICLR 2024, arXiv:2309.17425
>
> 2. > … since this paper uses OpenMaMMUT which involves not only CLIP-style contrastive learning but also sentence generation, it would be beneficial to show examples of the kinds of captions generated for various images …
>
> We agree that it will be a good reminder that openMaMMUT is also a functional captioner by providing examples of caption generation. Main focus of the study was to demonstrate the capability of our method to compare pure contrastive CLIP with contrastive+captioning loss mixture of MaMMUT, and check where scaling law based comparison can identify which procedure is the more scalable in this fully controlled setting. Thus, adding captioning loss served the purpose of demonstrating comparison between two controlled learning procedures, and while we gather evidence that contrastive+captioning loss mixture is what makes openMaMMUT perform better at larger scales on various downstream tasks compared to CLIP, we do not assume strong function of captioning itself. However, it is indeed helpful to present captioning examples nevertheless. In our additional experiments, we see examples of good captioning, eg correctly identifying multiple objects on the image or performing simple OCR from written text, as well as examples of failure, for instance not being able to parse calligraphic writing. We will update the revised manuscript with corresponding material and also provide a openMammut based demo for captioning on HuggingFace Spaces to try captioning on any dropped image.
>
> 3. > Similar contributions such as Molmo & PixMo (CVPR 2025) aim to open-source VLMs … discussing their positioning and relationship would enhance paper's overall value …
>
> We have a different focus than works like Molmo & PixMo, in 1) describing a novel generic method for robust comparison of various learning procedures (models & datasets) across scale span, while the mentioned work presents a single model and dataset showing performance on few selected scale points 2)  dealing with fully controlled scenario of single stage pre-training from scratch without relying on already pre-trained components, which makes training procedure and comparison fully reproducible, in contrast to Molmo which eg uses pre-trained language model and pre-trained vision encoder (with data for vision encoder being closed, making whole procedure not fully controllable and reproducible). In follow up works,  an interesting direction would be to apply scaling law based comparison to VLM post-training setting with fully open pre-trained components like studied in our work. This would enable VLM training procedure comparison that takes properly into account all stages, without having closed components (models & datasets) introducing non-controllable confounds. We will extend discussion in revised manuscript with the outlook to future work on scaling laws based comparison for multi stage VLM learning procedures with pre-trained components.
>
> 4. > … pre-training approaches … specialized for zero-shot classification, retrieval, and segmentation? If the underlying vision-language tasks share common characteristics, focusing on contrastive learning alone could suffice …
>
> In the study, we look at CLIP as a model using contrastive loss only and at MaMMUT as a model using contrastive+captioning loss to provide a compelling showcase for comparing two learning procedures using our method based on scaling law derivation across different settings. The comparison allows us to predict which of the procedures provides better performance at larger scales. It turns out to be MaMMUT that shows stronger scalability and outperforms CLIP at larger scales. Using our comparison method, we can thus decide whether contrastive learning alone could suffice to perform strongly on  those downstream tasks, which turns out to be not the case – as according to scaling law derivation based comparison, contrastive+captioning loss mixture performs consistently better across all the downstream tasks from certain compute threshold on (while on smaller scales on the other hand, contrastive learning via CLIP prevails), and thus poses the better, stronger scalable alternative to contrastive learning alone.

---

> > ### Comment · Reviewer_jP4c · 2025-08-07
> > **Official Comment by Reviewer jP4c**
> >
> > 1.
> > > We will release the pretrained models ... from all points measured for scaling law derivation
> >
> > Thank you so much for the great effort to the research community. The reviewer is also looking forward to the codes/models release.
> >
> >
> > 2.
> > >We will update the revised manuscript with corresponding material and also provide a openMammut based demo for captioning on HuggingFace Spaces to try captioning on any dropped image.
> >
> > The reviewer thanks the consideration and thinks it is enough to treat the comment.
> >
> > 3.
> > > We will extend discussion in revised manuscript with the outlook to future work on scaling laws based comparison for multi stage VLM learning procedures with pre-trained components.
> >
> > This is a simple comment, so the additional discussion is addressing the comment.

---

> ### Comment · Area_Chair_MZyL · 2025-08-05
>
> hi reviewer, thank you for spending time to read the paper and provide feedback. The authors have provided responses to your feedback and please spend some time to read the responses. Please do make sure let the authors know whether you still have any concerns, questions, or suggestions after reading their responses.

---

> ### Author Response · Authors · 2025-08-07
>
> We thank the reviewer for timely engagement and are glad to see our comments were helpful. The anonymous repository in the link https://anonymous.4open.science/r/open_clip_scaling-76D2/README.md (that was already part of the original submission) contains the code to reproduce the plots from the raw data, the collected raw data itself and also the patches that can be applied to publicly available open source openCLIP and CLIP Benchmark versions which allows executing all the experiments reported in the paper including training openMaMMUT. The full version will contain all the full source code necessary for reproducing whole experimental pipelines involved in scaling law measurements and derivation, including training experiments and model evaluation.

---

### Official Review · Reviewer_TkVk · 2025-07-03

**Clarity:** 3
**Significance:** 2
**Originality:** 2
**Rating:** 5
**Confidence:** 4

**Summary:**

The paper derives compute vs performance scaling laws for two open-source language vision procedures CLIP and MaMMUT on two open datasets, DataComp and Re-LAION. They release all the pre-trained models with their intermediate checkpoints and develop their openMaMMUT that achieves 80.3% accuracy on zero-shot ImageNet 1k. When comparing the two models, they use a really dense measurement with $|M| = 15$ model configurations and $|D|=11$ samples configurations. Their contributions are:

- The comparison is fully reproducible and the compute is measured in a dense manner.
- They find MaMMUT has better scalability than CLIP.
- Constant learning rate can recover the same trend with less compute.
- Release all checkpoints, which contribute to the open source community.

**Questions:**

1. In line 216, $\alpha$ is set to be negative. According to equation (1), it should be positive. Compared with the results in Figure 5, the paper might set $\alpha$ to be $-\alpha$ by mistake.

2. Line 62 and Line 186 is conflicting with each other. After looking at the appendix figure 6,7, I found line 62 is correct. This is one of the conclusions of the paper, so please confirm which one is correct.

3. In Figure 8, the paper shows the comparison of SigLIP, CoCa, CLIP, and MaMMUT on the DataComp-1.4B dataset. Everywhere else, the paper also compared the Re-LAION-1.4B dataset. Does the same pattern hold on Re-LAION-1.4B dataset?

4. I wonder whether the paper can be extended to more general questions, such as why MaMMUT has better scalability than CLIP. This can provide more insights into the fundamental question of how to design a more scalable model. Specifically, one can do a study on whether it is the training loss or the model architecture that makes MaMMUT more scalable. This can make the paper more generalizable, and add great significance to the paper.

**Ethical Concerns:**

["NO or VERY MINOR ethics concerns only"]

**Final Justification:**

I now understand that the focus of the paper is more on how to perform robust, reproducible comparison of training procedures for open foundation models, rather than solely related to the comparison of several models and datasets in the paper. The rigorous experiments and procedures can be referenced for later experiments and offer valuable guidance. I appreciate the new experiment from the authors for rebuttal. I thank the authors for their clarification that the MaMMUT vs Cap experiment helps to identify that the contrastive and captioning loss mixture creates a more scalable training procedure. I have also raised my score as the author clarified the general question the paper is answering and provided answers to my questions.

**Limitations:**

Yes

**Paper Formatting Concerns:**

No paper formatting concerns.

**Quality:**

4

**Strengths And Weaknesses:**

The paper has excellent quality, conducted experiments in a dense manner, used significant computing resources, and released all the pre-trained models with their intermediate checkpoints. All the experiments are conducted in a rigorous and reproducible manner. For clarity, there are some typos and inconsistencies mentioned below in the Question section. In terms of significance, I recognize the paper contributes to the open source community in terms of a systematic practice for scaling laws comparison, and release of both checkpoints and open source models. The analysis of the learning rate is also very helpful. However, the question investigated in the paper is specific to the two models and the two datasets. I think more general questions are needed to be investigated, and the details are posted in the Question section. For originality, the paper extends the scaling laws methodology for model selection, and also has its derivative and confidence interval analysis for the power law equation. The conceptual contribution is similar to the original scaling law paper.

---

> ### Author Rebuttal · Authors · 2025-07-27
>
> We thank the reviewer for time dealing thoroughly with our work. We appreciate the reviewer’s remarks on the excellent quality of the paper, emphasis on rigorous and reproducible experiment execution and on contributions to the open-source community. In following, we would like to address the relevant points raised by the reviewer:
>
> 1. Thanks for careful reading and catching the typos. At line 216, we will remove the incorrect minus sign for $\alpha$, reading "($\alpha=0.208$ vs. $\alpha=0.354$)". At line 186, Re-LAION-1.4B will be replaced by DataComp-1.4B, reading "... stronger scaling for DataComp-1.4B on classification ..."
>
> 2. >  the question investigated in the paper is specific to the two models and the two datasets …
>
> The general question our work addresses is how to perform robust, reproducible comparison of training procedures for open foundation models, both with regard to architectures and datasets involved. To showcase how such comparison based on scaling law derivation can be conducted, we choose CLIP and MaMMUT and open datasets DataComp-1.4B and Re-LAION-1.4B as an example study subjects – the posed question and method are though not specific to the models and datasets we took in our example. Already in this study, we look at further models, deriving scaling laws for comparison of CLIP and MaMMUT with SigLIP (App. Fig. 8 (a)), CoCa (App. Fig. 8 (b)), CaP (App. Fig. 17). For the rebuttal, using the same method, we conduct additional experiments on another important open reference dataset, DFN-1.4B [1], obtaining further model and dataset comparison on DFN-1.4B. We again obtain same trends from scaling laws derived on DFN-1.4B as observed for DataComp (Fig. 1) and Re-LAION (Fig. 2) – MaMMUT being stronger scalable than CLIP, providing further support for consistency of the comparison across datasets and downstream tasks. Dataset comparison reveals DFN-1.4B [1] leading to better training for both CLIP and MaMMUT than DataComp and Re-LAION. This shows that our method is generic and can be applied for robust model and dataset comparison in various scenarios. We will update the revised manuscript with corresponding material.
>
> [1] Data Filtering Networks. Fang et al, ICLR 2024, arXiv:2309.17425
>
> 3. > In Figure 8, the paper shows the comparison of SigLIP, CoCa, CLIP, and MaMMUT on the DataComp-1.4B dataset. Everywhere else, the paper also compared the Re-LAION-1.4B dataset. Does the same pattern hold on Re-LAION-1.4B dataset?
>
> Large fraction of available compute budget for the study was used to obtain measurements for the main comparison for CLIP and Mammut, deriving scaling laws and showing consistent trends across datasets and downstream tasks  (Fig. 1, 2). Across datasets, we executed 4604 measurements (full training runs) to derive scaling laws for both models, see summary table below:
>
> | Dataset   | Model Type | Points |
> |-------------------|-------------|--------|
> | **DataComp-1.4B** | CLIP        | 672    |
> |                   | MaMMUT      | 1010   |
> | **Re-LAION-1.4B** | CLIP        | 703    |
> |                   | MaMMUT      | 750    |
> | **DFN-1.4B**        | CLIP        | 732    |
> |                   | MaMMUT      | 737    |
>
>
> As compute cost for executing dense measurements for scaling law derivation is substantial (see App. Tab. 4), the budget was not sufficient to repeat the same extensive measurement procedure across datasets for further architectures, comparison for which was thus restricted to DataComp-1.4B. To show consistent patterns for other datasets, we conduct for the rebuttal additionally for the main comparison showcase in our study - CLIP vs MaMMUT - comparison of data efficiency on Re-LAION-1.4B, which confirms observations on DataComp. and complement this with another data efficiency comparison for DFN-1.4B. Both confirm the patterns observed on DataComp - MaMMUT being a more data efficient model than CLIP. Together with the additional experiments on DFN-1.4B described above, we can confirm the same pattern of consistent comparison outcomes for CLIP vs MaMMUT, pointing to MaMMUT as stronger scalable and more data efficient learning procedure than CLIP across all studied open datasets. We will update the revised manuscript with corresponding material.
>
> 4. > why MaMMUT has better scalability than CLIP … fundamental question of how to design a more scalable model … study on whether it is the training loss or the model architecture that makes MaMMUT more scalable.
>
> Evidence obtained from comparisons in our work allows us to draw conclusions about what makes MaMMUT scale stronger than other measured architectures. On the one hand, we have a comparison to CLIP, which has in contrast to MaMMUT only contrastive loss, thus ablating captioning loss and cross-attention from vision to text tower. On the other hand, we also compare to Cap (see App. Fig. 17 and App. Sec. E “Scaling behavior of other architectures”; for rebuttal, we additionally perform comparison MaMMUT vs Cap using log likelihood evaluation based on logprobs of the text tower), which has captioning loss and cross-attention, but no contrastive loss, thus ablating for contrastive loss. As both CLIP and Cap underperform MaMMUT especially at larger scales, we can conclude that neither contrastive loss nor captioning loss + cross-attention alone are responsible for providing stronger scalability, and it is the contrastive and captioning loss mixture which creates more scalable training procedure (see also App. Sec. E.).
>
> In our study, the comparisons between the various models are performed in a common reference frame (eg same open datasets) and across scales, while fully controlling for adding and removing components of the learning procedure for each scaling law derivation. This allows us to identify the effects of various ablations on learning procedure scalability. This is in contrast to other works, where on the one hand, unknown confounds like closed datasets make it hard or impossible to infer which of the differences between the learning procedures (algorithmic or dataset based) led to better performance, while on the other hand performance is measured only on few selected scales without estimating scaling trends, making unclear whether comparison holds for wider span at larger scales. Designing better learning procedures requires an approach to conduct guided search for more scalable procedures given the existing ones as reference baselines. This approach requires a robust method for training procedure comparison valid across scales that can operate in the same reference frame to control for any interventions - including controlling the datasets which requires open data, - and this is where our study makes progress.  We will update the revised manuscript with corresponding material and make our conclusions about why MaMMUT is a more scalable learning procedure clearer in main text’s discussion section.

---

> > ### Comment · Reviewer_TkVk · 2025-08-04
> >
> > Thank you for the detailed reply. I now understand that the focus of the paper is more on how to perform robust, reproducible comparison of training procedures for open foundation models, rather than solely related to the comparison of several models and datasets in the paper. The rigorous experiments and procedures can be referenced for later experiments and offer valuable guidance. I appreciate the new experiment from the authors for rebuttal. I thank the authors for their clarification that the MaMMUT vs Cap experiment helps to identify that the contrastive and captioning loss mixture creates a more scalable training procedure.
> >
> > I hope to see more discussion and a stronger conclusion on why MaMMUT is a more scalable learning procedure in the main text’s discussion section in the final edition. I have also raised my score as the author clarified the general question the paper is answering and provided answers to my questions.

---

> > > ### Author Response · Authors · 2025-08-04
> > >
> > > We are glad our reply and experiments helped to clarify the questions and thank the reviewer for engaging and raising the score. We will update the manuscript accordingly with additional material from the rebuttal.

---

### Official Review · Reviewer_Nr5N · 2025-07-03

**Clarity:** 3
**Significance:** 3
**Originality:** 3
**Rating:** 5
**Confidence:** 4

**Summary:**

This paper presents a rigorous and large-scale empirical study that uses scaling laws as a principled framework for comparing vision-language foundation models and datasets. The authors move beyond traditional single-point comparisons by deriving full scaling laws for two important architectures, CLIP (contrastive) and MaMMUT (contrastive + captioning), across a wide range of model sizes and data scales. The experiments are conducted on two large, open datasets, DataComp-1.4B and Re-LAION-1.4B.

The core finding is that MaMMUT, which incorporates a generative captioning loss, consistently demonstrates better scalability than the purely contrastive CLIP, meaning its performance improves more rapidly with increased compute. This trend is shown to be robust across different downstream tasks (classification, retrieval, segmentation) and training datasets. The paper also provides a valuable methodology for deriving these scaling laws, including techniques for handling constant learning rate schedules to reduce compute cost. Finally, the authors release all models, checkpoints, and code, culminating in a new SOTA open model, OpenMaMMUT-L/14, which validates the predictions of their scaling analysis.

**Questions:**

Is it possible that the chosen hyperparameter search space or the final selected parameters favor the MaMMUT setup? For instance, the presence of a captioning loss might act as a regularizer, making the model more stable during training with certain learning rates that would be unstable for CLIP alone. Without a more detailed report on the hyperparameter tuning process and sensitivities, a skeptic could argue that the CLIP baseline might not have been trained under its own optimal conditions, thus artificially enhancing MaMM-UT's apparent scaling advantage.

**Ethical Concerns:**

["NO or VERY MINOR ethics concerns only"]

**Final Justification:**

The author's response addressed almost all of my concerns, and I am willing to maintain my “accept” rating.

**Limitations:**

Yes

**Quality:**

3

**Strengths And Weaknesses:**

Strengths:
1. The paper's main contribution is to advocate for and demonstrate a more scientific and robust methodology for comparing foundation models. In an era where "SOTA" claims are often based on single-point evaluations with murky training details, this work provides a blueprint for principled comparison across scales. This is a significant and much-needed contribution to the community.
2. The sheer scale of the empirical study is a major strength. The authors have conducted an exhaustive number of pre-training runs, covering multiple model sizes (S/32 to H/14), extensive data scales (1.28M to 3B samples), two model architectures (CLIP, MaMMUT), and two large datasets (DataComp, Re-LAION). This dense grid of measurements allows for highly reliable scaling law fits.
3. The commitment to reproducibility is exemplary. By using open-source models (OpenCLIP, and their new OpenMaMMUT), open datasets, and promising to release all code, data, and checkpoints, the authors set a high standard. This work is not just a study; it is a valuable and verifiable public resource. The release of OpenMaMMUT-L/14 with 80.3% ImageNet-1k accuracy is a strong practical outcome.
4. The central result—that MaMMUT out-scales CLIP—is demonstrated consistently and robustly across different evaluation tasks, datasets, and even learning rate schedules. The "crossover" effect, where CLIP is better at small scales but MaMMUT overtakes it at larger scales, is a particularly insightful finding that would be missed by single-point comparisons.

Weaknesses:
The core comparison is between only two architectural paradigms: CLIP (contrastive-only) and MaMMUT (contrastive + generative). While this is a valuable comparison, the paper's title and claims suggest a more general framework for comparing "Open Foundation Language-Vision Models." The study does not include other important architectural variants, such as (1) SigLIP, which uses a different loss function (sigmoid loss) and has shown very strong performance, (2) BLIP and BLIP2, which are generative large Language-Vision Models, (3) LLAVA, QwenVL, etc which incorporate CLIP encoder with large LLMs.

---

> ### Author Rebuttal · Authors · 2025-07-28
>
> We thank the reviewer for time dealing thoroughly with our work. We appreciate the reviewer’s high appraisal of the study, some of the summaries are so precise and well written that we consider adapting it for our own communication. In following, we would like to address the relevant points raised by the reviewer:
>
> 1. > … The core comparison is between only two architectural paradigms: CLIP and MaMMUT … the paper's title and claims suggest a more general framework for comparing "Open Foundation Language-Vision Models."
>
> The general question our work addresses is how to perform robust, reproducible comparison of training procedures for open foundation models across scales, both with regard to architectures and datasets involved. We choose CLIP and MaMMUT and open datasets DataComp-1.4B and Re-LAION-1.4B as an example study showcasing how our method works for comparing important vision-language learning procedures. The method is though not specific to the chosen study subjects (and is also not restricted to language-vision scenario only) and can be applied to compare any learning procedures that feature open foundation models and open datasets by conducting scaling law derivation as described in our work.
>
> 2. > The study does not include other important architectural variants, such as (1) SigLIP, which uses a different loss function (sigmoid loss) and has shown very strong performance …
>
> We would like to clarify that we do look at SigLIP in our study, deriving scaling law on DataComp-1.4B for this architecture, where softmax in CLIP is replaced by sigmoid transfer function, and comparing with original CLIP architecture (App. Sec. E, “Scaling behavior of other architectures”). By executing controlled comparison on same open dataset DataComp 1.4B we obtain evidence that replacing softmax with sigmoid transfer function does not provide any benefit over CLIP, as we do not see any difference in scaling trends obtained from scaling laws derived on same open dataset (App. Fig. 8 (a), App. Sec. E). Original SigLIP study featured a closed dataset (WebLI), such that independent controlled comparison with CLIP there is in contrast not possible. Our result shows that strong performance reported for SigLIP previously is highly likely due to the closed dataset WebLI, and not due to replacing softmax with sigmoid transfer function as claimed in the original work.
>
>
> > 3. … (2) BLIP and BLIP2, which are generative large Language-Vision Models, (3) LLAVA, QwenVL, etc which incorporate CLIP encoder with large LLM …
>
> While methods from our study can be applied to compare various foundation models and datasets, we have deliberately chosen a scenario where 1) pretraining is happening from scratch from randomly initialized models, excluding confounds bound to taking already pre-trained components 2) pretraining procedure is simple single stage . This allows for fully controlled setting to perform scaling law derivation, providing same reference frame for the comparison of learning procedures. Learning procedures that lead to BLIP3, Qwen-VL and other similar works are using various pre-trained components, often trained under unknown conditions and performing training in various multiple stages, which necessitates further hyperparam and dataset decisions per stage. This is in contrast to our showcase, where all components are fully controlled, open and hyperparams can be tuned in the single stage, allowing to derive scaling law for the full extent of a given learning procedure to serve as base for validated, reproducible comparison to other procedures. Another consideration is available compute budget – each comparison requires scaling law derivation and we have chosen to use the available compute budget to demonstrate our method in clear way for comparison of important reference vision-language learning procedures CLIP and MaMMUT. We hope that our method can be used to compare other important learning procedures in search of more scalable open foundation models in follow up works, taking additional necessary compute required to execute such comparisons.
>
> > 4. Is it possible that the chosen hyperparameter search space or the final selected parameters favor the MaMMUT setup? … a skeptic could argue that the CLIP baseline might not have been trained under its own optimal conditions, thus artificially enhancing MaMMUT's apparent scaling advantage.
>
> For our main comparison showcase, CLIP vs MaMMUT, we used wide intervals to scan for both models hyperparams extensively. We provide here overview for the hyperparam intervals and resulting scans for both models on datasets used in the study (for rebuttal, we also performed experiments on another important open dataset, DFN-1.4B):
>
> | Model  | **Learning Rate** |       | **Global Batch Size** |       | **Warmup Steps** |       |
> |--------------------|------------------|-------|----------------------|-------|-----------------|-------|
> |                    | min              | max   | min                  | max   | min             | max   |
> | CLIP               | 1e-04           | 1e-02 | 512                  | 181 248 | 1000         | 15 000 |
> | MaMMUT             | 1e-05           | 1e-02 | 256                  | 181 632 | 50            | 20 000 |
>
>
> | Dataset   | Model | Points |
> |-------------------|-------------|--------|
> | **DataComp-1.4B** | CLIP        | 672    |
> |                   | MaMMUT      | 1010   |
> | **Re-LAION-1.4B** | CLIP        | 703    |
> |                   | MaMMUT      | 750    |
> | **DFN-1.4B**        | CLIP        | 732    |
> |                   | MaMMUT      | 737    |
>
> As evident from the table, the measurements are balanced for Re-LAION (750 vs 703 points), as well as for DFN (737 vs 732 points). For DataComp, we have 672 scans for CLIP and 1010 scans for MaMMUT on DataComp-1.4B. More measurements were performed on DataComp for MaMMUT to avoid unfair advantage for CLIP, as CLIP architecture was already extensively tuned in numerous previous works, which is not the case for MaMMUT that was not open sourced before and was not available for tuning by the broad community as was the case for eg openCLIP.  Confirming there is no tuning advantage, predictions from scaling laws on held out points from DataComp have same high accuracy and low uncertainty for both CLIP and MaMMUT, also showing similar RMSE  - App. Tab. 7, here a short version:
>
> | Model             | Samples Seen | GFLOPs    | IN1k 0-shot acc |  Predicted IN1k 0-shot acc (95% CI) |
> |------------------|--------------|-----------|:------------------:|:--------------------------------------------------:|
> | **CLIP**                                                                                                                                            |
> | ViT-H-14         | 3.07e+9      | 1.14e+12  | 0.784            | 0.779 (0.770, 0.789)                             |
> | *RMSE: 5.90e-03*                                                                                                    |
> | **MaMMUT**                                                                                                                                         |
> | mammut-ViT-H-14  | 3.07e+9      | 1.43e+12  | 0.798    | 0.801 (0.793, 0.809)                             |
> | *RMSE: 7.57e-03*                                                                                                    |
>
> The predictions are accurate using held-out points on compute-optimal Pareto front for both CLIP and MaMMUT, as we see actual measurements on compute-optimal Pareto front falling well into the 95% confidence interval for the predictions on held-out points. (For the fit, points up to $C^{MaMMUT}_{threshold} = 2.59\cdot 10^{11}$ GFLOPS for MaMMUT and up to
>
> $C^{CLIP}_{threshold} = 4.07\cdot 10^{11}$ GFLOPS for CLIP were taken, which extrapolates for MaMMUT 5.5 and for CLIP 2.8 times; see App. Tab. 7). Similar high prediction accuracy for both models again hints on absence of tuning advantage for either CLIP or MaMMUT.
>
> Further, we conduct an additional experiment to double check whether there is any difference in obtained scaling law if working with the same number of points for MaMMUT as for CLIP on DataComp-1.4B. We perform bootstrapping, sampling randomly 672 points from 1010 available points for MaMMUT, doing 10 trials, fitting scaling law for each trial and averaging the obtained scaling law coefficients. We observe that the obtained fit coefficients have no significant difference from scaling law obtained with 1000 points:
>
> | Model                                           | Ac        | Bc        | αc        | Ec        |
> |------------------------------------------------|-----------|-----------|-----------|-----------|
> | CLIP                                            | 57.862083 | 18.391321 | -0.226604 | 0.111169  |
> | MaMMUT (full points)                            | 125.356572| 19.289384 | -0.255670 | 0.101112  |
> | MaMMUT (same points num. as CLIP, 10 random samples) | 125.267163| 19.301461 | -0.255934 | 0.108208  |
>
> Thus, comparison on reduced points shows again the same trends. Further, comparison via scaling laws on Re-LAION (Fig. 2) and DFN (will be included in the revised manuscript) which have balanced number of scanned points gives same trends as on DataComp, showing MaMMUT stronger scalable than CLIP, again confirming no influence of tuning on comparison. We can thus conclude from the gathered evidence that observed trends and performed comparison are valid and are due to differences in learning procedure (MaMMUT having contrastive+captioning loss mixture vs contrastive only for CLIP), and cannot be due to differences in tuning. We will update the revised manuscript with corresponding material.

---

> > ### Comment · Reviewer_Nr5N · 2025-08-06
> >
> > Thank you for your response. I have carefully read your rebuttal and the comments from other reviewers. I am pleased that some of my concerns have been addressed, so I have decided to maintain my positive rating.

---

> ### Author Response · Authors · 2025-08-06
>
> We thank the reviewer for timely engagement and are glad to hear that our response was helpful to solve concerns.
>
> As reviewer expressed specifically interest in impact of tuning on observed performance across scales and datasets, we provide additional summary across selection of reference scales, showing actual measurements from the performed experiments for the 3 datasets on the Pareto front of best tuned models, together with their tuned hyperparams. We see that Mammut consistently outperforms CLIP, while tuned hyperparams for both often have same values. This provides further evidence that tuning for both converges to very similar hyperparam regions and does not give advantage or disadvantage for any of the studied architectures.
>
> IN1K downstream performance (the "-" entries correspond to missing measurements, where also comparison is not possible)
>
> | Dataset           | Model Scale   | Samples Scale |  CLIP  | MaMMUT | Diff. (Mammut benefit) |
> |-------------------|--------------|-------------------:|-------:|-------:|------------------------:|
> | **Re-LAION-1.4B** | **ViT-B-32** | 307M               | 0.4401 | 0.4574 | +0.0173 |
> |                   |              | 640M               | 0.5122 | 0.5181 | +0.0059 |
> |                   |              | 1B                 | 0.5601 | 0.5634 | +0.0033 |
> |                   |              | 3B                 | 0.6065 | 0.6075 | +0.0010 |
> |                   | **ViT-B-16** | 307M               | 0.5104 | 0.5434 | +0.0330 |
> |                   |              | 640M               | 0.5810 | 0.6094 | +0.0284 |
> |                   |              | 1B                 | 0.6257 | 0.6439 | +0.0182 |
> |                   |              | 3B                 | 0.6677 | 0.6830 | +0.0153 |
> |                   | **ViT-L-14** | 307M               | 0.5662 | 0.6020 | +0.0358 |
> |                   |              | 640M               | 0.6365 | 0.6652 | +0.0287 |
> |                   |              | 1B                 | 0.6818 | 0.7061 | +0.0243 |
> |                   |              | 3B                 | 0.7206 | 0.7369 | +0.0163 |
> |                   | **ViT-H-14** | 307M               | 0.5898 | 0.6214 | +0.0316 |
> |                   |              | 640M               | 0.6575 | 0.6843 | +0.0268 |
> |                   |              | 1B                 | 0.7029 | 0.7258 | +0.0229 |
> |                   |              | 3B                 | -      | 0.7615 | -       |
>
>
> | Dataset       | Model Scale | Samples Scale |  CLIP  | MaMMUT | Diff. (Mammut benefit) |
> |---------------|------------|-------------------:|-------:|-------:|-----------:|
> | **DataComp-1.4B** | **ViT-B-32** | 307M | 0.5028 | 0.5257 | +0.0229 |
> |               |            | 640M | -    | 0.5852 | -     |
> |               |            | 1B   | 0.6076 | 0.6209 | +0.0133 |
> |               |            | 3B   | 0.6529 | 0.6613 | +0.0084 |
> |               | **ViT-B-16** | 307M | 0.5777 | 0.6148 | +0.0371 |
> |               |            | 640M | 0.6405 | 0.6693 | +0.0288 |
> |               |            | 1B   | 0.6679 | 0.6908 | +0.0229 |
> |               |            | 3B   | 0.7133 | 0.7267 | +0.0134 |
> |               | **ViT-L-14** | 307M | 0.6323 | 0.6685 | +0.0362 |
> |               |            | 640M | 0.6987 | 0.7240 | +0.0253 |
> |               |            | 1B   | 0.7317 | 0.7488 | +0.0171 |
> |               |            | 3B   | 0.7656 | 0.7845 | +0.0189 |
> |               | **ViT-H-14** | 307M | 0.6483 | 0.6912 | +0.0429 |
> |               |            | 640M | 0.7178 | 0.7410 | +0.0232 |
> |               |            | 1B   | 0.7517 | 0.7682 | +0.0165 |
> |               |            | 3B   | 0.7840 | 0.7986 | +0.0146 |
>
> | Dataset       | Model Scale   | Samples Scale |  CLIP  | MaMMUT | Diff. (Mammut benefit) |
> |---------------|--------------|-------------------:|-------:|-------:|------------------------:|
> | **DFN-1.4B**  | **ViT-B-32** | 307M               | 0.5524 | 0.5682 | +0.0158 |
> |               |              | 640M               | 0.6104 | 0.6218 | +0.0114 |
> |               |              | 1B                 | 0.6402 | 0.6599 | +0.0197 |
> |               |              | 3B                 | 0.6943 | 0.6947 | +0.0004 |
> |               | **ViT-B-16** | 307M               | 0.6274 | 0.6516 | +0.0242 |
> |               |              | 640M               | 0.6779 | 0.6991 | +0.0212 |
> |               |              | 1B                 | 0.7059 | 0.7268 | +0.0209 |
> |               |              | 3B                 | -      | 0.7541 | -       |
> |               | **ViT-L-14** | 307M               | 0.6777 | 0.7021 | +0.0244 |
> |               |              | 640M               | 0.7258 | 0.7470 | +0.0212 |
> |               |              | 1B                 | 0.7609 | 0.7777 | +0.0168 |
> |               | **ViT-H-14** | 307M               | 0.6955 | 0.7190 | +0.0235 |
> |               |              | 640M               | 0.7477 | 0.7666 | +0.0189 |

---

> > ### Author Response · Authors · 2025-08-06
> >
> > Tuned hyperams for best models on compute optimal Pareto front for IN1K downstream performance for Re-LAION, DataComp, and DFN (performed for the rebuttal), showing high similarity for both CLIP and MaMMUT (the "-" entries correspond to training not performed on corresponding combination of scales)
> >
> >
> > | Dataset          | Model Size   | Samples Seen | Learning Rate |        | Warmup |        | Batch Size |        |
> > |------------------|--------------|-------------:|--------------:|-------:|-------:|-------:|-----------:|-------:|
> > |                  |              |              | CLIP          | MaMMUT | CLIP   | MaMMUT | CLIP       | MaMMUT |
> > | **Re-LAION-1.4B**| **ViT-B-32** | 307M         | 2.00e-03      | 2.00e-03 | 4.00e+03 | 4.00e+03 | 1.64e+04 | 1.64e+04 |
> > |                  |              | 640M         | 2.00e-03      | 2.00e-03 | 4.00e+03 | 6.00e+03 | 3.28e+04 | 3.28e+04 |
> > |                  |              | 1B           | 2.00e-03      | 1.50e-03 | 6.00e+03 | 6.00e+03 | 6.55e+04 | 3.28e+04 |
> > |                  |              | 3B           | 2.00e-03      | 2.00e-03 | 6.00e+03 | 6.00e+03 | 9.01e+04 | 9.01e+04 |
> > |                  | **ViT-B-16** | 307M         | 1.50e-03      | 1.50e-03 | 4.00e+03 | 4.00e+03 | 1.64e+04 | 1.64e+04 |
> > |                  |              | 640M         | 2.00e-03      | 2.00e-03 | 6.00e+03 | 4.00e+03 | 4.51e+04 | 4.51e+04 |
> > |                  |              | 1B           | 2.00e-03      | 2.00e-03 | 6.00e+03 | 6.00e+03 | 6.55e+04 | 6.55e+04 |
> > |                  |              | 3B           | 2.00e-03      | 2.00e-03 | 1.00e+04 | 6.00e+03 | 6.55e+04 | 9.01e+04 |
> > |                  | **ViT-L-14** | 307M         | 2.00e-03      | 2.00e-03 | 3.00e+03 | 3.00e+03 | 3.28e+04 | 3.28e+04 |
> > |                  |              | 640M         | 2.00e-03      | 2.00e-03 | 4.00e+03 | 4.00e+03 | 4.51e+04 | 4.51e+04 |
> > |                  |              | 1B           | 4.00e-03      | 2.00e-03 | 6.00e+03 | 4.00e+03 | 9.01e+04 | 9.01e+04 |
> > |                  |              | 3B           | 2.00e-03      | 2.00e-03 | 4.00e+03 | 4.00e+03 | 9.01e+04 | 9.01e+04 |
> > |                  | **ViT-H-14** | 307M         | 2.00e-03      | 1.50e-03 | 4.00e+03 | 4.00e+03 | 3.28e+04 | 3.28e+04 |
> > |                  |              | 640M         | 2.00e-03      | 1.50e-03 | 4.00e+03 | 4.00e+03 | 4.51e+04 | 4.51e+04 |
> > |                  |              | 1B           | 2.00e-03      | 1.50e-03 | 6.00e+03 | 6.00e+03 | 9.01e+04 | 9.01e+04 |
> > |                  |              | 3B           | -             | 4.00e-03 | -      | 6.00e+03 | -         | 1.80e+05 |
> >
> >
> >
> > | Dataset          | Model Size   | Samples Seen | Learning Rate |        | Warmup |        | Batch Size |        |
> > |------------------|--------------|-------------:|--------------:|-------:|-------:|-------:|-----------:|-------:|
> > |                  |              |              | CLIP          | MaMMUT | CLIP   | MaMMUT | CLIP       | MaMMUT |
> > | **DataComp-1.4B**| **ViT-B-32** | 307M         | 2.00e-03      | 2.00e-03 | 4.00e+03 | 4.00e+03 | 1.64e+04 | 1.64e+04 |
> > |                  |              | 640M         | -             | 1.50e-03 | -      | 4.00e+03 | -         | 1.64e+04 |
> > |                  |              | 1B           | 1.00e-03      | 2.00e-03 | 1.50e+04 | 4.00e+03 | 1.64e+04 | 1.64e+04 |
> > |                  |              | 3B           | 3.00e-03      | 2.00e-03 | 4.00e+03 | 4.00e+03 | 9.06e+04 | 4.53e+04 |
> > |                  | **ViT-B-16** | 307M         | 2.00e-03      | 2.00e-03 | 4.00e+03 | 4.00e+03 | 1.64e+04 | 1.64e+04 |
> > |                  |              | 640M         | 2.00e-03      | 2.00e-03 | 6.00e+03 | 4.00e+03 | 3.28e+04 | 3.28e+04 |
> > |                  |              | 1B           | 1.00e-03      | 1.00e-03 | 4.00e+03 | 4.00e+03 | 1.64e+04 | 1.64e+04 |
> > |                  |              | 3B           | 2.00e-03      | 2.00e-03 | 4.00e+03 | 4.00e+03 | 4.53e+04 | 9.06e+04 |
> > |                  | **ViT-L-14** | 307M         | 1.50e-03      | 1.50e-03 | 3.00e+03 | 4.00e+03 | 3.28e+04 | 1.64e+04 |
> > |                  |              | 640M         | 2.00e-03      | 2.00e-03 | 4.00e+03 | 6.00e+03 | 4.51e+04 | 4.51e+04 |
> > |                  |              | 1B           | 4.00e-03      | 2.00e-03 | 4.00e+03 | 4.00e+03 | 9.06e+04 | 9.06e+04 |
> > |                  |              | 3B           | 4.00e-03      | 2.00e-03 | 4.00e+03 | 4.00e+03 | 9.11e+04 | 9.11e+04 |
> > |                  | **ViT-H-14** | 307M         | 2.00e-03      | 1.50e-03 | 6.00e+03 | 6.00e+03 | 1.63e+04 | 1.63e+04 |
> > |                  |              | 640M         | 2.00e-03      | 2.00e-03 | 4.00e+03 | 6.00e+03 | 4.51e+04 | 3.20e+04 |
> > |                  |              | 1B           | 2.00e-03      | 2.00e-03 | 6.00e+03 | 6.00e+03 | 9.01e+04 | 9.02e+04 |
> > |                  |              | 3B           | 1.50e-03      | 1.50e-03 | 4.00e+03 | 6.00e+03 | 9.02e+04 | 9.02e+04 |

---

> > > ### Author Response · Authors · 2025-08-06
> > >
> > > | Dataset      | Model Size   | Samples Seen | Learning Rate |        | Warmup |        | Batch Size |        |
> > > |--------------|--------------|-------------:|--------------:|-------:|-------:|-------:|-----------:|-------:|
> > > |              |              |              | CLIP          | MaMMUT | CLIP   | MaMMUT | CLIP       | MaMMUT |
> > > | **DFN-1.4B** | **ViT-B-32** | 307M         | 1.50e-03      | 2.00e-03 | 4.00e+03 | 4.00e+03 | 1.64e+04 | 1.64e+04 |
> > > |              |              | 640M         | 1.00e-03      | 2.00e-03 | 6.00e+03 | 6.00e+03 | 1.64e+04 | 3.28e+04 |
> > > |              |              | 1B           | 2.00e-03      | 2.00e-03 | 6.00e+03 | 6.00e+03 | 9.01e+04 | 6.55e+04 |
> > > |              |              | 3B           | 2.00e-03      | 2.00e-03 | 1.00e+04 | 1.00e+04 | 6.55e+04 | 6.55e+04 |
> > > |              | **ViT-B-16** | 307M         | 2.00e-03      | 1.50e-03 | 4.00e+03 | 4.00e+03 | 1.64e+04 | 1.64e+04 |
> > > |              |              | 640M         | 2.00e-03      | 2.00e-03 | 4.00e+03 | 6.00e+03 | 4.51e+04 | 3.28e+04 |
> > > |              |              | 1B           | 1.00e-03      | 1.50e-03 | 6.00e+03 | 6.00e+03 | 1.64e+04 | 3.28e+04 |
> > > |              |              | 3B           | -             | 2.00e-03 | -      | 6.00e+03 | -         | 9.01e+04 |
> > > |              | **ViT-L-14** | 307M         | 2.00e-03      | 2.00e-03 | 4.00e+03 | 4.00e+03 | 3.28e+04 | 1.64e+04 |
> > > |              |              | 640M         | 2.00e-03      | 2.00e-03 | 6.00e+03 | 6.00e+03 | 3.28e+04 | 4.51e+04 |
> > > |              |              | 1B           | 4.00e-03      | 2.00e-03 | 4.00e+03 | 4.00e+03 | 9.01e+04 | 9.01e+04 |
> > > |              | **ViT-H-14** | 307M         | 1.50e-03      | 1.50e-03 | 4.00e+03 | 4.00e+03 | 3.28e+04 | 1.64e+04 |
> > > |              |              | 640M         | 2.00e-03      | 2.00e-03 | 4.00e+03 | 4.00e+03 | 4.51e+04 | 4.51e+04 |
> > >
> > >
> > > We will update the manuscript accordingly with additional material and thank again for the discussion that was helpful to further clarify and improve the work.

---

### Comment · Area_Chair_MZyL · 2025-08-02
**Author-reviewer discussion**

Dear Authors and Reviewers,

I would like to thank the authors for providing detailed rebuttal messages.

For the **reviewers**, I would like to encourage you to carefully read all other reviews and the author responses and engage in an open exchange with the authors. Please post your first response as soon as possible within the discussion time window, so there is time for back and forth discussion with the authors. All reviewers should respond to the authors, so that the authors know their rebuttal has been read.

Cheers, AC

---

### Note · Authors · 2025-08-14

Dear reviewers and chairs,

we thank for the highly constructive and engaged review process. As finalization remark, we provide a hopefully helpful summary of rebuttal discussions from our perspective.

1. Scaling laws based open foundation models and datasets comparison

Our work presents a method for robust comparison of learning procedures via scaling law derivation in a fully controlled, reproducible setting. Reviewers pointed out that derived scaling laws are novel on their own (Reviewers jP4c, Nr5N), as we provide for the first time densely derived scaling laws for open datasets like DataComp and Re-LAION (experiments on DFN added during rebuttal) for both CLIP and MaMMUT. Reviewers emphasize that main contribution is to demonstrate “a more scientific and robust methodology for comparing foundation models”, pointing out this being “a significant and much-needed contribution to the community”, “in an era where SOTA claims are often based on single-point evaluations with murky training details”, with our work providing “a blueprint for principled comparison across scales” (Reviewer Nr5N). Reviewers also noted that extending scaling law methodology to allow more accurate predictions and for providing confidence intervals to estimate prediction uncertainty is original and aids the robust comparison (Reviewer TkVk). Reviewer Mi8q commented on the appropriate rigor given the extensive balanced hyperparameter tuning for the performed comparison. We are glad to note that, also according to the opinion of the reviewers, conducting robust scaling law-based comparisons of architectures and datasets under open, reproducible conditions addresses a critical gap in current evaluation methods, moving beyond single-point comparisons to systematic, scale-aware assessment which can also be used by the community as a common open reference to compare to and improve upon.

2. openMaMMUT as strong model obtained by following comparison’s predictions, open-source release.

We were glad to see that reviewers appreciated openMaMMUT as strong outcome and further validation of the performed comparison procedure. We would like to note again that full release following publication will contain all the trained models used for scaling law derivation, training source code, model evaluation and scaling law fitting procedure source code (main parts included in submission's anonymous repo), making it easy to check, fully reproduce (also due to open datasets) and build upon the study.

---

### Decision · Program_Chairs · 2025-09-17

**Decision:**

Accept (poster)

**Comment:**

The paper presents a scaling law study on CLIP (contrastive) and MaMMUT (contrastive + captioning) architectures with  DataComp-1.4B and Re-LAION-1.4B datasets. The scaling law shows that MaMMUT consistently demonstrates better scalability than the purely contrastive CLIP, where MaMMUT performance improves more rapidly with increased compute, on a diverse downstream tasks (classification, retrieval, segmentation) and training datasets. The authors also release models, checkpoints, and code.

Strength of the paper:
- Working on a popular and important area (vision language representation learning)
- Reasonable scale of experiments and number of different configurations to support generalizability of findings
- Generic experiment design and good selection of downstream evaluation tasks provides useful knowledge for a wide range of readers
- Due to the scale of the investigation (dataset and model size etc.) , it's hard to reproduce experiments for academic community anyway. However, releasing the best model obtained in the investigation can facilitate many practitioners in their works, and open sourcing the implementation code etc. still encourage the community to be more open.

Reviewers align on most of the strengths of the paper from the high level and see its novelty, contribution, and relevance to NeurIPS venue. Thus we recommend this paper to be accepted to the conference.

After most questions being addressed and clarified during the rebuttal and discussion stage, reviewers feel a few areas that can improve the paper.

To improve:
- Foundation Language-Vision Models is a wide area while this paper explored two examples (CLIP and MaMMUT) of the models. It's better to have a title better describing the work
- Many discussion between reviewers and authors are valuable to include in the final manuscript